# Registration is a Powerful Rotation-Invariance Learner for 3D Anomaly Detection

**Yuyang Yu**[1*]    **Zhengwei Chen**[1*]    **Xuemiao Xu**[1†]    **Lei Zhang**[2†]    **Haoxin Yang**[1]
**Yongwei Nie**[1]    **Shengfeng He**[3]

[1]South China University of Technology    [2]Guangdong University of Petrochemical Technology
[3]Singapore Management University

yyyoung0611@gmail.com    chen_zw_123@163.com
{xuemx, nieyongwei}@scut.edu.cn    harxis@outlook.com
zhanglei@gdupt.edu.cn    shengfenghe@smu.edu.sg
https://github.com/CHen-ZH-W/Reg2Inv

## Abstract

3D anomaly detection in point-cloud data is critical for industrial quality control, aiming to identify structural defects with high reliability. However, current memory bank-based methods often suffer from inconsistent feature transformations and limited discriminative capacity, particularly in capturing local geometric details and achieving rotation invariance. These limitations become more pronounced when registration fails, leading to unreliable detection results. We argue that point-cloud registration plays an essential role not only in aligning geometric structures but also in guiding feature extraction toward rotation-invariant and locally discriminative representations. To this end, we propose a registration-induced, rotation-invariant feature extraction framework that integrates the objectives of point-cloud registration and memory-based anomaly detection. Our key insight is that both tasks rely on modeling local geometric structures and leveraging feature similarity across samples. By embedding feature extraction into the registration learning process, our framework jointly optimizes alignment and representation learning. This integration enables the network to acquire features that are both robust to rotations and highly effective for anomaly detection. Extensive experiments on the Anomaly-ShapeNet and Real3D-AD datasets demonstrate that our method consistently outperforms existing approaches in effectiveness and generalizability.

## 1 Introduction

3D anomaly detection aims to identify structural defects in point-cloud data, with critical applications in industrial quality control. Despite the emergence of domain-specific datasets such as Real3D-AD [1] and Anomaly-ShapeNet [2], the rarity of real-world anomalies (e.g., orientation shift and other irregularities [3–5]) presents a persistent challenge. Consequently, most existing approaches adopt unsupervised paradigms that rely solely on normal samples for training.

Among unsupervised approaches, memory bank-based methods have shown promise by maintaining a repository of normal features and computing anomaly scores based on their deviation from incoming test samples. To address spatial misalignment between test samples and stored prototypes, several recent methods [1, 6, 7] universally adopt FPFH [8] with RANSAC-based [9] coarse registration prior to feature extraction. However, as illustrated in Figure 1(a), significant residual misalignments often persist even after registration. While registration failures harm anomaly detection performance,

---

* The first two authors contributed equally.
† Corresponding authors.

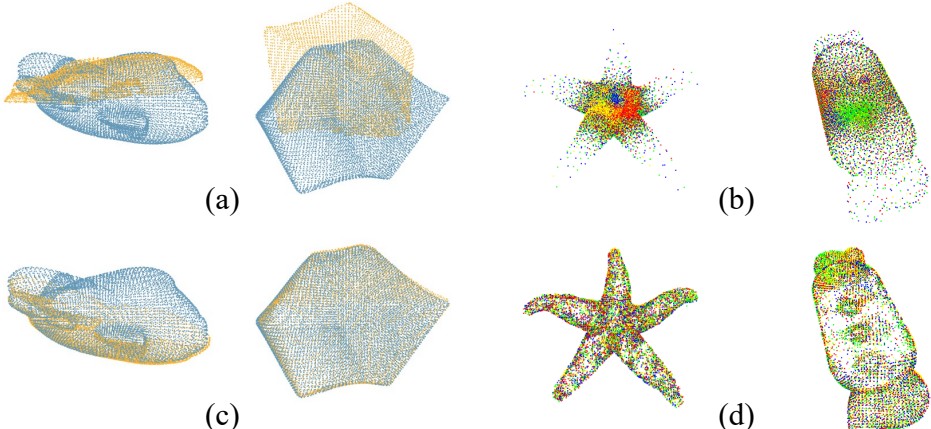

Figure 1: (a) Existing methods still exhibit misalignments after registration. (b) Visualization of PointMAE features in the memory bank, with different colors for different objects. (c) Our method achieves accurate point cloud alignment. (d) Visualization of our locally discriminative features in the memory bank.

it also points to a deeper problem: the feature encoders themselves lack the geometric sensitivity and transformation consistency required to support reliable anomaly detection.

This limitation stems largely from the nature of commonly used encoders such as PointMAE [10], which are designed to capture global semantics but often fail to preserve fine-grained local geometry(Figure 1(b)) and essential invariances. In particular, these encoders are not optimized to be rotation-invariant or to capture local structural nuances, both of which are essential for effective anomaly detection. As a result, feature representations struggle to maintain consistent correspondences under varying orientations and structural perturbations, especially in the presence of anomalies. When registration errors occur, these weaknesses exacerbate feature misalignment, ultimately resulting in unreliable anomaly scores.

To address these challenges, we propose a shift in how registration is employed within the anomaly detection pipeline. Rather than using it as a separate preprocessing module, we treat registration as an integral component of feature learning. Our key insight is that both point cloud registration and memory-based anomaly detection rely on the same core capabilities: modeling local geometric structures and capturing meaningful feature similarities across samples. Registration, by design, requires learning features that are rotation-invariant, locally sensitive, and structurally discriminative to establish accurate correspondences between source and target point clouds. Similarly, anomaly detection depends on features that can preserve fine-grained local details and remain consistent under rigid transformations, enabling precise comparisons between a test sample and normal prototypes stored in a memory bank. By aligning feature learning with registration objectives, our approach naturally yields representations well-suited for anomaly detection, overcoming the limitations of conventional encoders like PointMAE in handling geometric variations and structural defects.

Building on this insight, we propose a unified framework, called **Reg2Inv**, to derive *registration-induced rotation invariance* for 3D anomaly detection. During training, the model learns features through a registration task that enforces both geometric alignment and multi-scale feature consistency between source and target point clouds. This process not only establishes accurate structural correspondences but also shapes the feature extractor to produce rotation-invariant and locally discriminative representations(Figure 1(d)). At inference time, the model extracts features from a test point cloud, computes a registration matrix to align it with a prototype, and then compares the normalized features to those in a coreset-sampled memory bank. Anomaly scores are derived from these comparisons, allowing the system to identify local defects even under rotation or deformation. By jointly optimizing feature learning and alignment, our method effectively addresses spatial misalignment(Figure 1(c)) and enhances feature robustness, enabling more reliable 3D anomaly detection. Experimental results on Anomaly-ShapeNet and Real3D-AD confirm the superiority of our approach over existing baselines.

In summary, our contributions are threefold:

- We delve into the intrinsic alignment between point cloud registration and memory bank-based 3D anomaly detection, and show that registration is a powerful feature learner for acquiring discriminative and rotation-invariant representations.

- We propose a unified framework Reg2Inv for registration-induced rotation-invariant feature extraction, which jointly performs accurate prototype-sample alignment and enables effective anomaly scoring based on robust local geometric features.

- We conduct extensive experiments on two benchmark datasets, Anomaly-ShapeNet and Real3D-AD, demonstrating that our method consistently outperforms existing state-of-the-art approaches in 3D anomaly detection.

## 2 Related Work

### 2.1 2D Anomaly Detection

Anomaly detection in 2D images has seen significant progress in recent years, particularly under anomaly-free training settings. Popular approaches include feature embedding methods, among which flow-based, memory bank-based, and reconstruction-based techniques are widely adopted. Flow-based methods [11–14] model the distribution of normal features using normalizing flows and detect anomalies via likelihood estimation. Memory bank-based approaches [15–20] store features from pre-trained encoders and identify anomalies by comparing test samples to stored normal patterns. Reconstruction-based methods [21–26] learn to reconstruct normal inputs and detect anomalies through reconstruction errors. In this work, we focus on anomaly detection in 3D point clouds. Unlike structured 2D images, point clouds are unstructured, unordered, and often sparse, posing greater challenges for feature learning and anomaly detection.

### 2.2 3D Anomaly Detection

3D anomaly detection targets structural irregularities in point-cloud data [27, 28] and has seen increasing research attention. Existing methods generally fall into two categories: reconstruction-based and memory bank-based. Among reconstruction-based methods, IMRNet [2] detects anomalies by reconstructing masked normal samples; R3D-AD [29] restores normal geometry from pseudo-abnormal inputs; and PO3AD [30] enhances local reconstruction by predicting offsets in defective regions. While these methods are effective at capturing fine-grained anomalies, they often suffer from sensitivity to resolution and noise.

Memory bank-based methods aim to learn compact representations of normal structures for comparison. Reg3D-AD [1] uses PointMAE [10] to extract features and stores both features and coordinates in separate memory banks. Group3AD [6] introduces a group-level feature aggregation strategy to improve anomaly sensitivity and alignment. ISMP [7] proposes an internal spatial modality perception framework that leverages a spatial insight engine for enhanced feature discrimination. While these methods have shown promising results, they often overlook the limitations of current registration strategies and, more importantly, reflect a deeper issue: feature representations frequently lack transformation consistency and local discriminability. In contrast, our work integrates point cloud registration into the anomaly detection pipeline to jointly optimize spatial alignment and feature learning. By aligning feature extraction with registration objectives, our framework generates rotation-invariant and locally discriminative features, significantly enhancing detection robustness.

### 2.3 Coarse-to-Fine Point Cloud Registration

Coarse-to-fine strategies emerge as a compelling approach to capture hierarchical visual structure [31, 32], which also has been proven to be effective in both 2D image matching [33–35] and 3D point cloud registration [36, 37]. Geometric Transformer [37], in particular, achieves robust registration by incorporating geometric structure into attention-based models. Our approach builds upon this line of work by not only employing a coarse-to-fine registration pipeline but also enhancing it with feature learning objectives that ensure both robust alignment and discriminative local feature extraction.

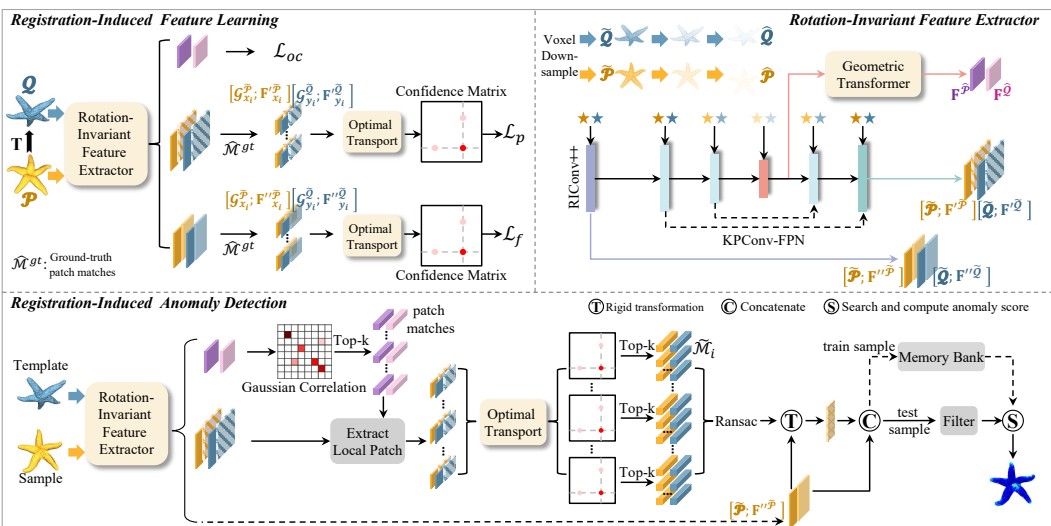

Figure 2: The overview of Reg2Inv. In the feature learning stage, the model learns features via a registration task that enforces geometric alignment and multi-scale consistency between paired point clouds. At inference time, the model computes a registration matrix for alignment with the prototype and extracts rotation-invariant features for anomaly detection. The feature extractor is designed to obtain a set of features for anomaly detection and two sets of features for registration.

## 3 Methodology

**Problem statement.** The 3D anomaly detection task involves a training set $\mathcal{D}_{\text{train}} = \{\mathcal{P}_i^n \in \mathbb{R}^{N_i \times 3}\}_{i=1}^M$ containing $M$ normal objects and a test set $\mathcal{D}_{\text{test}} = \{\mathcal{P}_i^n \in \mathbb{R}^{N_i \times 3}\}_{i=1}^J \cup \{\mathcal{P}_i^a \in \mathbb{R}^{N_i \times 3}\}_{i=1}^K$, consisting of $J$ normal and $K$ abnormal objects (where $n$ and $a$ denote normal and abnormal, respectively). Each normal object $\mathcal{P}^n$ contains only normal points $p^n$, while an abnormal object $\mathcal{P}^a$ includes both $p^n$ and abnormal points $p^a$. The goal of this task is to train models on $\mathcal{D}_{\text{train}}$ with two purposes: (1) object-level anomaly detection: distinguishing $\mathcal{P}^n$ and $\mathcal{P}^a$ in $\mathcal{D}_{\text{test}}$; and (2) point-level anomaly localization: identify $p^a$ within $\mathcal{P}^a$ to localize anomalies.

**Overview.** As shown in Figure 2, our framework **Reg2Inv** consists of two stages: *Registration-Induced Feature Learning* and *Registration-Induced Anomaly Detection*, both built upon a *Rotation-Invariant Feature Extractor*. The model performs point-cloud registration in the feature learning stage by enforcing geometric and feature consistency across multi-scale patches and points. This refines the feature extractor to learn rotation-invariant and discriminative features while aligning structural correspondences. During anomaly detection, the model extracts rotation-invariant features from test samples, computes a registration matrix to align them with a prototype, and identifies anomalies by comparing normalized features to a coreset-sampled memory bank.

### 3.1 Registration-Induced Feature Learning

**Point sampling & Ground-truth matches generation.** In the feature learning stage, we first generate transformed pairs $(\mathcal{P}, \mathcal{Q})$ by applying random rigid transformations $T_{\text{gt}}$ to each $\mathcal{P} \in \mathcal{D}_{\text{train}}$, yielding $\mathcal{Q}$. Since both point cloud registration and anomaly detection benefit from reduced data complexity, and dense point clouds lead to redundant or invalid point-wise alignments, we employ multi-scale voxel downsampling on both $\mathcal{P}$ and $\mathcal{Q}$, obtaining the first level points $\tilde{\mathcal{P}}$, $\tilde{\mathcal{Q}}$ and the coarsest level points $\hat{\mathcal{P}}$, $\hat{\mathcal{Q}}$. Notably, $\hat{\mathcal{P}}$ and $\hat{\mathcal{Q}}$ are voxel downsampled from $\tilde{\mathcal{P}}$ and $\tilde{\mathcal{Q}}$, their relationships can be captured via a point-to-node grouping strategy [36–38]. Specifically, each $\tilde{p} \in \tilde{\mathcal{P}}$ is associated with its nearest coarsest-level point $\hat{p} \in \hat{\mathcal{P}}$, forming patches $\mathcal{G}^{\mathcal{P}}$. The construction of $\mathcal{G}^{\mathcal{Q}}$ follows the same procedure applied to $\tilde{\mathcal{Q}}$ and $\hat{\mathcal{Q}}$. Formally, the patch $\mathcal{G}^{\mathcal{P}}$ is defined as:

$$\mathcal{G}_i^{\mathcal{P}} = \{\tilde{p} \in \tilde{\mathcal{P}} \mid i = argmin_j (\|\tilde{p} - \hat{p}_j\|_2), \hat{p}_j \in \hat{\mathcal{P}}\}. \tag{1}$$

Given the $\mathcal{G}^{\mathcal{P}}$ and $\mathcal{G}^{\mathcal{Q}}$, we construct two types of ground-truth matches between them to enable supervised loss computation. Specifically, under the known rigid transformation $T_{\text{gt}}$, we compute the

overlap ratio between patch pairs $(\mathcal{G}_i^{\mathcal{P}}, \mathcal{G}_{\tilde{j}}^{\mathcal{Q}})$. Patch pairs with an overlap ratio $> 0.1$ are selected as ground-truth patch matches:

$$\hat{\mathcal{M}}^{\text{gt}} = \{(\mathcal{G}_i^{\mathcal{P}}, \mathcal{G}_{\tilde{j}}^{\mathcal{Q}}) \mid \text{Overlap}(T_{\text{gt}}(\mathcal{G}_i^{\mathcal{P}}), \mathcal{G}_{\tilde{j}}^{\mathcal{Q}}) > 0.1\}. \tag{2}$$

In addition, we randomly sample $N_g$ patch pairs from $\hat{\mathcal{M}}^{\text{gt}}$ and establish ground-truth point matches by confidence threshold $t$ as follows:

$$\tilde{\mathcal{M}}_i^{\text{gt}} = \{(\tilde{p}, \tilde{q}) \mid \|T_{\text{gt}}(\tilde{p}) - \tilde{q}\|_2 < t, \ \tilde{p} \in \mathcal{G}_{x_i}^{P}, \ \tilde{q} \in \mathcal{G}_{y_i}^{Q}\}, \tag{3}$$

where $(\mathcal{G}_{x_i}^{\mathcal{P}}, \mathcal{G}_{y_i}^{\mathcal{Q}}) \in \hat{\mathcal{M}}^{\text{gt}}$ and $|\tilde{\mathcal{M}}^{\text{gt}}| = N_g$.

**Features extraction.** To leverage registration guidance for extracting discriminative and rotation-invariant features tailored to anomaly detection, we design a feature extractor comprising three components: RIConv++ [39], a KPConv-FPN backbone [40, 41], and a Geometric Transformer module [37] with layer normalization. To capture fine-grained geometric details from $\tilde{\mathcal{P}}$ and $\tilde{\mathcal{Q}}$, we apply RIConv++ to extract dense local features features $\mathbf{F}''^{\tilde{\mathcal{P}}} \in \mathbb{R}^{|\tilde{\mathcal{P}}| \times d''}$ and $\mathbf{F}''^{\tilde{\mathcal{Q}}} \in \mathbb{R}^{|\tilde{\mathcal{Q}}| \times d''}$, where $d''$ denotes the local feature dimension. These features encode rotation-invariant and geometrically coherent local structures, which are essential for anomaly detection. To encode hierarchical information across multiple resolutions, we employ the KPConv-FPN to process multi-scale point clouds and generate multi-level point-wise features. The corresponding outputs for $\tilde{\mathcal{P}}$ and $\tilde{\mathcal{Q}}$ are denoted as $\mathbf{F}'^{\tilde{\mathcal{P}}} \in \mathbb{R}^{|\tilde{\mathcal{P}}| \times d'}$ and $\mathbf{F}'^{\tilde{\mathcal{Q}}} \in \mathbb{R}^{|\tilde{\mathcal{Q}}| \times d'}$, where $d'$ denotes the point feature dimension. These features capture discriminative contextual patterns at the point level, enabling accurate and robust spatial alignment. Finally, $\hat{\mathcal{P}}$ and $\hat{\mathcal{Q}}$ are processed using the Geometric Transformer module with layer normalization, yielding patch-wise features $\mathbf{F}^{\hat{\mathcal{P}}} \in \mathbb{R}^{|\hat{\mathcal{P}}| \times d}$ and $\mathbf{F}^{\hat{\mathcal{Q}}} \in \mathbb{R}^{|\hat{\mathcal{Q}}| \times d}$, where $d$ is the final patch feature dimension. These features represent global structural priors that facilitate reliable registration under partial overlap or sparsity.

Given these extracted features, for each patch $\mathcal{G}_i^{\mathcal{P}}$, we use $\hat{\mathcal{M}}^{\text{gt}}$ to define its local feature matrix and point feature matrix as $\mathbf{F}''^{\tilde{\mathcal{P}}}_i \subset \mathbf{F}''^{\tilde{\mathcal{P}}}$ and $\mathbf{F}'^{\tilde{\mathcal{P}}}_i \subset \mathbf{F}'^{\tilde{\mathcal{P}}}$, respectively. The local feature matrix $\mathbf{F}''^{\tilde{\mathcal{Q}}}_j$ and point feature matrix $\mathbf{F}'^{\tilde{\mathcal{Q}}}_j$ for each patch $\mathcal{G}_j^{\mathcal{Q}}$ are computed and denoted in a similar way.

**Training objective** To achieve accurate point cloud registration while simultaneously leveraging the registration process to extract rotation-invariant features suitable for anomaly detection, our overall training objective is formulated as a combination of three complementary components:

$$\mathcal{L} = \mathcal{L}_f + \mathcal{L}_p + \mathcal{L}_{oc}. \tag{4}$$

Here, $\mathcal{L}_f$ is a **negative log-likelihood loss** [42] used for aligning local features. This loss is key in encouraging the network to learn geometrically coherent and rotation-invariant representations, which are crucial for effective anomaly detection. For each ground-truth patch match $(\mathcal{G}_{x_i}^{\mathcal{P}}, \mathcal{G}_{y_i}^{\mathcal{Q}}) \in \hat{\mathcal{M}}^{\text{gt}}$, we first employ the optimal transport layer [42] to extract a local feature assignment matrix, from which we then compute the corresponding cost matrix $\mathcal{C}''_i$. Subsequently, we augment each cost matrix $\mathcal{C}''_i$ by adding one row and one column filled with a learnable dustbin parameter $\alpha$, forming $\mathcal{C}''^*_i$. We then utilize the Sinkhorn [43] algorithm on $\mathcal{C}''^*_i$ to obtain a soft local feature assignment matrix $Z''^*_i$. The $Z''^*_i$ is used to compute the $\mathcal{L}_{f,i}$ defined as:

$$\mathcal{L}_{f,i} = -\sum_{(x,y) \in \tilde{\mathcal{M}}_i^{\text{gt}}} \log z''^*_{i,x,y} - \sum_{x \in \mathcal{I}_i} \log z''^*_{i,x,m_i+1} - \sum_{y \in \mathcal{J}_i} \log z''^*_{i,n_i+1,y}. \tag{5}$$

where $\tilde{\mathcal{M}}_i^{\text{gt}}$ represents the set of matched points, while $\mathcal{I}_i$ and $\mathcal{J}_i$ denote the sets of unmatched points in $\mathcal{G}_{x_i}^{\mathcal{P}}$ and $\mathcal{G}_{y_i}^{\mathcal{Q}}$. The $\mathcal{L}_f$ is computed by averaging the individual loss: $\mathcal{L}_f = \frac{1}{N_g} \sum_{i=1}^{N_g} \mathcal{L}_{f,i}$.

$\mathcal{L}_p$ and $\mathcal{L}_{oc}$ are used to supervise point cloud registration at different levels of granularity. $\mathcal{L}_p$ follows a negative log-likelihood loss and shares a similar computation process with $\mathcal{L}_f$. It is tailored for point matching, refining point-level alignment, and enforcing accurate correspondences between individual points, which improves cross-cloud matching accuracy and enhances registration robustness.

$\mathcal{L}_{oc}$ denotes the **overlap-aware circle loss** [37, 44], which prioritizes correspondences in regions with significant overlap. This helps improve the reliability of registration under partial overlap or sparse input by focusing on structurally consistent patches. Its definition is as follows:

$$\mathcal{L}_{oc}^{\mathcal{P}} = \frac{1}{|\mathcal{A}|} \sum_{\mathcal{G}_i^{\mathcal{P}} \in \mathcal{A}} \log \left[ 1 + \sum_{\mathcal{G}_j^{\mathcal{Q}} \in \varepsilon_p^i} e^{\lambda_i^j \beta_p^{i,j} \left( d_i^j - \Delta_p \right)} \cdot \sum_{\mathcal{G}_k^{\mathcal{Q}} \in \varepsilon_n^i} e^{\beta_n^{i,k} \left( \Delta_n - d_i^k \right)} \right]. \tag{6}$$

The synergy among these losses ensures accurate and robust registration, while promoting rotation-invariant and geometrically coherent representations, which are crucial for reliable anomaly detection. Further details on the loss formulations are provided in the appendix.

## 3.2 Registration-Induced Anomaly Detection

**Point alignment.** During the inference phase, a sample from $\mathcal{D}_{\text{train}}$ is selected as the template $\mathcal{Q}$, and all samples in both $\mathcal{D}_{\text{train}}$ and $\mathcal{D}_{\text{test}}$ must undergo registration to align with this template. The same point sampling and feature extraction from training were applied to all samples. To find patch matches of each sample pair $(\mathcal{P}, \mathcal{Q})$, we compute a Gaussian correlation matrix $\mathcal{H} \in \mathbb{R}^{|\hat{\mathcal{P}}| \times |\hat{\mathcal{Q}}|}$ with $h_{i,j} = \exp(-\|\mathbf{F}_i^{\hat{\mathcal{P}}} - \mathbf{F}_j^{\hat{\mathcal{Q}}}\|_2^2)$ as in [37]. We then perform a dual-normalization operation [35, 45] on $\mathcal{H}$ to obtain an augmented correlation matrix $\bar{\mathcal{H}}$. Finally, we select the largest $N_c$ entries in $\bar{\mathcal{H}}$ as the patch matches:

$$\hat{\mathcal{M}} = \left\{ (\hat{\mathbf{p}}_{x_i}, \hat{\mathbf{q}}_{y_i}) \mid (x_i, y_i) \in \text{topk}_{x,y} \left( \bar{h}_{x,y} \right) \right\}. \tag{7}$$

For each match $(\mathcal{G}_{x_i}^{\mathcal{P}}, \mathcal{G}_{y_i}^{\mathcal{Q}}) \in \hat{\mathcal{M}}$, we compute its point assignment matrix $Z_i'^*$ which is then recovered to $Z_i'$ by dropping the last row and the last column. We select the largest $k$ entries in $Z_i'$ as the point matches:

$$\tilde{\mathcal{M}}_i = \left\{ \left( \mathcal{G}_{x_i}^{\mathcal{P}} (x_j), \mathcal{G}_{y_i}^{\mathcal{Q}} (y_j) \right) \mid (x_j, y_j) \in \text{mutual\_topk}_{x,y} (z'_{i,x,y}) \right\}. \tag{8}$$

The point matches computed from each patch match are then collected together to estimate the rigid transformation $T$ by RANSAC [9]. Subsequently, the estimated transformation $T$ is applied to align $\tilde{\mathcal{P}}$ with $\tilde{\mathcal{Q}}$:

$$\tilde{\mathcal{P}}^{align} = T(\tilde{\mathcal{P}}). \tag{9}$$

**Feature normalization and memory bank construction.** For each train sample $\mathcal{P} \in \mathcal{D}_{\text{train}}$, we derive local features $\mathbf{F}''^{\tilde{\mathcal{P}}} \in \mathbb{R}^{|\tilde{\mathcal{P}}| \times d''}$ and the corresponding alignment coordinates $\tilde{\mathcal{P}}^{align} \in \mathbb{R}^{|\tilde{\mathcal{P}}| \times 3}$. These are then aggregated across all training samples in $\mathcal{D}_{\text{train}}$ to form collections of feature vectors $\mathbf{F}_f$ and coordinate vectors $\mathbf{F}_c$, from which we compute the normalization parameters $\gamma_f$ and $\gamma_c$, respectively. The final representation $\mathbf{F}$ is obtained by normalizing $\mathbf{F}_f$ and $\mathbf{F}_c$ using $\gamma_f$ and $\gamma_c$, and then fusing them via a concatenation-based operator, that is $\mathbf{F} = \Phi(\mathbf{F}_f/\gamma_f, \mathbf{F}_c/\gamma_c)$. Finally, we apply the Coreset sampling technique [1, 15, 46] to construct the memory bank $\mathcal{B}$.

**Feature filtering and anomaly detection.** For each test sample $\mathcal{P} \in \mathcal{D}_{\text{test}}$, we compute the final representation $\mathbf{F}$ in the same way. Feature filtering is performed to eliminate edge artifacts and ensure enhanced feature normalization across test samples. Specifically, we construct local neighborhoods around the feature-corresponding points by computing $\mathcal{N}_i = \text{KNN}(p_i, k)$, where $p_i \in \mathcal{P}$. We then calculate the centroid of each local neighborhood:

$$\mu_i = \frac{1}{|\mathcal{N}_i|} \sum_{p_j \in \mathcal{N}_i} p_j. \tag{10}$$

The feature filtering process can be summarized by the following equation:

$$\text{Filter}(i) = \begin{cases} 1 & \text{if } \|p_i - \mu_i\| = \min_{p \in \mathcal{N}_i} \|p - \mu_i\| \\ 0 & \text{otherwise} \end{cases}. \tag{11}$$

The filtered feature set $\mathbf{F}_{fil} = \{f_i \mid \text{Filter}(i) = 1\}$ is subsequently employed for anomaly detection. The point-level anomaly score $s_i$ is defined as:

$$s_i = \min_{f^{bank} \in \mathcal{B}} \|f_i - f^{bank}\|_2, \tag{12}$$

where $f_i \in \mathbf{F}_{fil}$. The object-level anomaly score $\mathcal{S}$ is computed by aggregating all point-level scores into a score mask and taking the maximum value after smoothing: $\mathcal{S} = \max(\{s_i\} * f_n)$, where $f_n$ is a mean filter of size $n$ and $*$ is the point-wise convolution operator.

Table 1: Comparison of AUROC results at the object and point levels (%) of various methods on Real3D-AD. The best result in red and the second-best in blue for each category. (Raw) denotes the raw point coordinates used as input to the method. (FPFH) and (PMAE) denote configurations using Fast Point Feature Histograms [8] and PointMAE [10] as feature extractors. The top three methods are reconstruction-based, while the remaining ones are memory bank-based.

| | O-AUROC(↑) / P-AUROC(↑) | | | | | | |
|---|---|---|---|---|---|---|---|
| Method | Airplane | Car | Candy | Chicken | Diamond | Duck | Fish |
| IMRNet | 76.2 / | 71.1 / | 75.5 / | 78.0 / | 90.5 / | 51.7 / | 88.0 / |
| R3D-AD | 77.2 / | 69.3 / | 71.3 / | 71.4 / | 68.5 / | 90.9 / | 69.2 / |
| PO3AD | 80.4 / | 65.4 / | 78.5 / | 68.6 / | 80.1 / | 82.0 / | 85.9 / |
| BTF(RAW) | 73.0 / 56.4 | 64.7 / 64.7 | 53.9 / 73.5 | 78.9 / 60.9 | 70.7 / 56.3 | 69.1 / 60.1 | 60.2 / 51.4 |
| BTF(FPFH) | 52.0 / 73.8 | 56.0 / 70.8 | 63.0 / 86.4 | 43.2 / 73.5 | 54.5 / 88.2 | 78.4 / 87.5 | 54.9 / 70.9 |
| M3DM | 43.4 / 54.7 | 54.1 / 60.2 | 55.2 / 67.9 | 68.3 / 67.8 | 60.2 / 60.8 | 43.3 / 66.7 | 54.0 / 60.6 |
| PatchCore(FPFH) | 88.2 / 56.2 | 59.0 / 75.4 | 54.1 / 78.0 | 83.7 / 42.9 | 57.4 / 82.8 | 54.6 / 26.4 | 67.5 / 82.9 |
| PatchCore(PMAE) | 72.6 / 56.9 | 49.8 / 60.9 | 66.3 / 62.7 | 82.7 / 72.9 | 78.3 / 71.8 | 48.9 / 52.8 | 63.0 / 71.7 |
| CPMF | 70.1 / 61.8 | 55.1 / 83.6 | 55.2 / 73.4 | 50.4 / 55.9 | 52.3 / 75.3 | 58.2 / 71.9 | 55.8 / 98.8 |
| Reg3D-AD | 71.6 / 63.1 | 69.7 / 71.8 | 68.5 / 72.4 | 85.2 / 67.6 | 90.0 / 83.5 | 58.4 / 50.3 | 91.5 / 82.6 |
| Group3AD | 74.4 / 63.6 | 72.8 / 74.5 | 84.7 / 73.8 | 78.6 / 75.9 | 93.2 / 86.2 | 67.9 / 63.1 | 97.6 / 83.6 |
| ISMP | 85.8 / 75.3 | 73.1 / 83.6 | 85.2 / 90.7 | 71.4 / 79.8 | 94.8 / 92.6 | 71.2 / 87.6 | 94.5 / 88.6 |
| Ours | 81.8 / 92.3 | 75.8 / 94.4 | 100. / 96.9 | 94.4 / 91.0 | 100. / 97.9 | 75.0 / 93.7 | 67.2 / 84.6 |

| Method | Gemstone | Seahorse | Shell | Starfish | Toffees | Average | |
|---|---|---|---|---|---|---|---|
| IMRNet | 67.4 / | 60.4 / | 66.5 / | 67.4 / | 77.4 / | 72.5 / | |
| R3D-AD | 66.5 / | 72.0 / | 84.0 / | 70.1 / | 70.3 / | 73.4 / | |
| PO3AD | 69.3 / | 75.6 / | 80.0 / | 75.8 / | 77.1 / | 76.5 / | |
| BTF(RAW) | 68.6 / 59.7 | 59.6 / 52.0 | 39.6 / 48.9 | 53.0 / 39.2 | 70.3 / 62.3 | 63.5 / 57.1 | |
| BTF(FPFH) | 64.8 / 89.1 | 77.9 / 51.2 | 75.4 / 57.1 | 57.5 / 50.1 | 46.2 / 81.5 | 60.3 / 73.3 | |
| M3DM | 64.4 / 67.4 | 49.5 / 56.0 | 69.4 / 73.8 | 55.1 / 53.2 | 45.0 / 68.2 | 55.2 / 63.1 | |
| PatchCore(FPFH) | 37.0 / 91.0 | 50.5 / 73.9 | 58.9 / 73.9 | 44.1 / 60.6 | 56.5 / 74.7 | 59.3 / 68.2 | |
| PatchCore(PMAE) | 37.4 / 44.4 | 53.9 / 63.3 | 50.1 / 70.9 | 51.9 / 58.0 | 58.5 / 58.0 | 59.4 / 62.0 | |
| CPMF | 58.9 / 44.9 | 72.9 / 96.2 | 65.3 / 72.5 | 70.0 / 80.0 | 39.0 / 95.9 | 58.6 / 75.8 | |
| Reg3D-AD | 41.7 / 54.5 | 76.2 / 81.7 | 58.3 / 81.1 | 50.6 / 61.7 | 82.7 / 75.9 | 70.4 / 70.5 | |
| Group3AD | 53.9 / 56.4 | 84.1 / 82.7 | 58.5 / 79.8 | 56.2 / 62.5 | 79.6 / 80.3 | 75.1 / 73.5 | |
| ISMP | 46.8 / 85.7 | 72.9 / 81.3 | 62.3 / 83.9 | 66.0 / 64.1 | 84.2 / 89.5 | 75.7 / 83.6 | |
| Ours | 73.5 / 90.7 | 53.2 / 64.5 | 69.2 / 90.6 | 84.1 / 84.0 | 62.6 / 73.7 | 78.0 / 87.8 | |

## 4 Experiments

### 4.1 Experimental Settings

**Datasets.** Evaluation is conducted on Anomaly-ShapeNet [2] and Real3D-AD [1]. Anomaly-ShapeNet is a synthetic 3D anomaly detection dataset with 1,600 samples across 40 categories, each containing 4 normal samples in the training set. Real3D-AD is a real-world high-resolution dataset with 12 object categories, each category has 4 normal training samples and 100 test instances. To simulate practical scenarios, training samples in Real3D-AD are captured via full $360°$ scans, while test samples are collected from single-view perspectives.

**Evaluation metrics.** We evaluate anomaly detection performance at both the object and point levels using the Area Under the Receiver Operating Characteristic Curve (AUROC). Object-level AUROC (O-AUROC) measures the effectiveness of anomaly detection, while Point-level AUROC (P-AUROC) assesses localization accuracy. Higher values in both metrics indicate stronger anomaly detection and localization capabilities.

**Details.** We set the local feature dimension to 32. For adaptive voxelization, the voxel size is dynamically determined via binary search to maintain 8,192 sampled points on Real3D-AD and 4,096 on Anomaly-ShapeNet. The model is trained for 100k iterations using the Adam optimizer with an initial learning rate of 1e-4. The learning rate is first linearly warmed up for 10k steps, then decayed following a cosine schedule to 10% of the initial value. Experiments are conducted on a single RTX 4090D GPU.

**Baselines.** Our method is evaluated against two categories of state-of-the-art 3D anomaly detection approaches on Anomaly-ShapeNet: (1) **Reconstruction-based methods**, including IMRNet [2], R3D-AD [29], and PO3AD [30]; and (2) **Memory bank-based methods**, including BTF [47],

Table 2: Comparison of AUROC results at the object and point levels (%) of various methods on Anomaly-ShapeNet.

| Method | ashtray0 | bag0 | bottle0 | bottle1 | bottle3 | bowl0 | bowl1 | bowl2 | bowl3 | bowl4 | bowl5 |
|---|---|---|---|---|---|---|---|---|---|---|---|
| | O-AUROC(↑) / P-AUROC(↑) | | | | | | | | | | |
| IMRNet | 67.1 / 67.1 | 66.0 / 66.8 | 55.2 / 55.6 | 70.0 / 70.2 | 64.0 / 64.1 | 68.1 / 78.1 | 70.2 / 70.5 | 68.5 / 68.4 | 59.9 / 59.9 | 67.6 / 57.6 | 71.0 / 71.5 |
| R3D-AD | 83.3 / | 72.0 / | 73.3 / | 73.7 / | 78.1 / | 81.9 / | 77.8 / | 74.1 / | 76.7 / | 74.4 / | 65.6 / |
| PO3AD | 100. / 96.2 | 83.3 / 94.9 | 90.0 / 91.2 | 93.3 / 84.4 | 92.6 / 88.0 | 92.2 / 97.8 | 82.9 / 91.4 | 83.3 / 91.8 | 88.1 / 93.5 | 98.1 / 96.7 | 84.9 / 94.1 |
| BTF(RAW) | 57.8 / 51.2 | 41.0 / 43.0 | 59.7 / 55.1 | 51.0 / 49.1 | 56.8 / 72.0 | 56.4 / 52.4 | 26.4 / 46.4 | 52.5 / 42.6 | 38.5 / 68.5 | 66.4 / 56.3 | 41.7 / 51.7 |
| BTF(FPFH) | 42.0 / 62.4 | 54.6 / 74.6 | 34.4 / 64.1 | 54.6 / 54.9 | 32.2 / 62.2 | 50.9 / 71.0 | 66.8 / 76.8 | 51.0 / 51.8 | 49.0 / 59.0 | 60.9 / 67.9 | 69.9 / 69.9 |
| M3DM | 57.7 / 57.7 | 53.7 / 63.7 | 57.4 / 66.3 | 63.7 / 63.7 | 54.1 / 53.2 | 63.4 / 65.8 | 66.3 / 66.3 | 68.4 / 69.4 | 61.7 / 65.7 | 46.4 / 62.4 | 40.9 / 48.9 |
| PatchCore(FPFH) | 58.7 / 59.7 | 57.1 / 57.4 | 60.4 / 65.4 | 66.7 / 68.7 | 57.2 / 51.2 | 50.4 / 52.4 | 63.9 / 53.1 | 61.5 / 62.5 | 53.7 / 32.7 | 49.4 / 72.0 | 55.8 / 35.8 |
| PatchCore(PMAE) | 59.1 / 49.5 | 60.1 / 67.4 | 51.3 / 55.3 | 60.1 / 60.6 | 65.0 / 65.3 | 52.3 / 52.7 | 62.9 / 52.4 | 45.8 / 51.5 | 57.9 / 58.1 | 50.1 / 50.1 | 59.3 / 56.2 |
| CPMF | 35.3 / 61.5 | 64.3 / 65.5 | 52.0 / 52.1 | 48.2 / 57.1 | 40.5 / 43.5 | 78.3 / 74.5 | 63.9 / 48.8 | 62.5 / 63.5 | 65.8 / 64.1 | 68.3 / 68.3 | 68.5 / 68.4 |
| Reg3D-AD | 59.7 / 69.8 | 70.6 / 71.5 | 48.6 / 88.6 | 69.5 / 69.6 | 52.5 / 52.5 | 67.1 / 77.5 | 52.5 / 61.5 | 49.0 / 59.3 | 34.8 / 65.4 | 66.3 / 80.0 | 59.3 / 69.1 |
| ISMP | / 60.3 | / 74.7 | / 77.0 | / 56.8 | / 77.5 | / 85.1 | / 54.6 | / 73.6 | / 77.3 | / 74.0 | / 53.4 |
| Ours | 90.0 / 78.5 | 100. / 99.1 | 100. / 99.5 | 100. / 84.9 | 100. / 81.7 | 100. / 98.3 | 80.7 / 82.8 | 65.6 / 82.2 | 58.5 / 76.1 | 85.2 / 78.8 | 81.8 / 82.4 |

| Method | bucket0 | bucket1 | cap0 | cap3 | cap4 | cap5 | cup0 | cup1 | eraser0 | headset0 | headset1 |
|---|---|---|---|---|---|---|---|---|---|---|---|
| IMRNet | 58.0 / 58.5 | 77.1 / 77.4 | 73.7 / 71.5 | 77.5 / 70.6 | 65.2 / 75.3 | 65.2 / 74.2 | 64.3 / 64.3 | 75.7 / 68.8 | 54.8 / 54.8 | 72.0 / 70.5 | 67.6 / 47.6 |
| R3D-AD | 68.3 / | 75.6 / | 82.2 / | 73.0 / | 68.1 / | 67.0 / | 77.6 / | 75.7 / | 89.0 / | 73.8 / | 79.5 / |
| PO3AD | 85.3 / 75.5 | 78.7 / 89.9 | 87.7 / 95.7 | 85.9 / 94.8 | 79.2 / 94.0 | 67.0 / 86.4 | 87.1 / 90.9 | 83.3 / 93.2 | 99.5 / 97.4 | 80.8 / 82.3 | 92.3 / 90.7 |
| BTF(RAW) | 61.7 / 61.7 | 32.1 / 68.6 | 66.8 / 52.4 | 52.7 / 68.7 | 46.8 / 46.9 | 37.3 / 37.3 | 40.3 / 63.2 | 52.1 / 56.1 | 52.5 / 63.7 | 37.8 / 57.8 | 51.5 / 47.5 |
| BTF(FPFH) | 40.1 / 40.1 | 63.3 / 63.3 | 61.8 / 73.0 | 52.2 / 65.8 | 52.0 / 52.4 | 58.6 / 58.6 | 58.6 / 79.0 | 61.0 / 61.9 | 71.9 / 71.9 | 52.0 / 62.0 | 49.0 / 59.1 |
| M3DM | 30.9 / 69.8 | 50.1 / 69.9 | 55.7 / 53.1 | 42.3 / 60.5 | 77.7 / 71.8 | 63.9 / 65.5 | 53.9 / 71.5 | 55.6 / 55.6 | 62.7 / 71.0 | 57.7 / 58.1 | 61.7 / 58.5 |
| PatchCore(FPFH) | 46.9 / 45.9 | 55.1 / 57.1 | 58.0 / 47.2 | 45.3 / 65.3 | 75.7 / 59.5 | 79.0 / 79.5 | 60.0 / 65.5 | 58.6 / 59.6 | 65.7 / 81.0 | 58.3 / 58.3 | 63.7 / 46.4 |
| PatchCore(PMAE) | 59.3 / 58.6 | 56.1 / 57.4 | 58.9 / 54.4 | 47.6 / 48.8 | 72.7 / 72.5 | 53.8 / 54.5 | 61.0 / 51.0 | 55.6 / 85.6 | 67.7 / 37.8 | 59.1 / 57.5 | 62.7 / 42.3 |
| CPMF | 48.2 / 48.6 | 60.1 / 60.1 | 60.1 / 60.1 | 55.1 / 55.1 | 55.3 / 55.3 | 69.7 / 55.1 | 49.7 / 49.7 | 49.9 / 50.9 | 68.9 / 68.9 | 64.3 / 69.9 | 45.8 / 45.8 |
| Reg3D-AD | 61.0 / 61.9 | 75.2 / 75.2 | 69.3 / 63.2 | 72.5 / 71.8 | 64.3 / 81.5 | 46.7 / 46.7 | 51.0 / 68.5 | 53.8 / 69.8 | 34.3 / 75.5 | 53.7 / 58.0 | 61.0 / 62.6 |
| ISMP | / 52.4 | / 67.2 | / 86.5 | / 73.4 | / 75.3 | / 67.8 | / 86.9 | / 60.0 | / 70.6 | / 58.0 | / 70.2 |
| Ours | 81.3 / 61.0 | 90.2 / 85.5 | 65.9 / 86.1 | 86.3 / 94.5 | 68.1 / 86.4 | 90.2 / 97.0 | 73.3 / 79.8 | 93.3 / 88.1 | 100. / 98.0 | 100. / 94.6 | 84.3 / 97.0 |

| Method | helmet0 | helmet1 | helmet2 | helmet3 | jar0 | phone | shelf0 | tap0 | tap1 | vase0 | vase1 |
|---|---|---|---|---|---|---|---|---|---|---|---|
| IMRNet | 59.7 / 59.8 | 60.0 / 60.4 | 64.1 / 64.4 | 57.3 / 66.3 | 78.0 / 76.5 | 75.5 / 74.2 | 60.3 / 60.5 | 67.6 / 68.1 | 69.6 / 69.9 | 53.3 / 53.5 | 75.7 / 68.5 |
| R3D-AD | 75.7 / | 72.0 / | 63.3 / | 70.7 / | 83.8 / | 76.2 / | 69.6 / | 73.6 / | 90.0 / | 78.8 / | 72.9 / |
| PO3AD | 76.2 / 87.8 | 96.1 / 94.8 | 86.9 / 93.2 | 75.4 / 84.6 | 86.6 / 87.1 | 77.6 / 81.0 | 57.3 / 66.3 | 74.5 / 78.3 | 68.1 / 69.2 | 85.8 / 95.5 | 74.2 / 88.2 |
| BTF(RAW) | 55.3 / 50.4 | 60.2 / 60.5 | 52.6 / 70.0 | 42.0 / 42.3 | 56.3 / 58.3 | 16.4 / 46.4 | 52.5 / 52.7 | 57.3 / 56.4 | 53.1 / 61.8 | 54.9 / 54.9 | |
| BTF(FPFH) | 57.1 / 57.5 | 71.9 / 74.9 | 54.2 / 64.3 | 44.4 / 72.4 | 42.4 / 42.7 | 67.1 / 67.5 | 60.9 / 61.9 | 56.0 / 56.8 | 54.6 / 59.6 | 34.2 / 64.2 | 21.9 / 61.9 |
| M3DM | 52.6 / 59.9 | 42.7 / 42.7 | 62.3 / 62.3 | 37.4 / 65.5 | 44.1 / 54.1 | 35.7 / 35.8 | 56.4 / 55.4 | 75.4 / 64.6 | 73.9 / 71.2 | 42.3 / 60.8 | 42.7 / 60.2 |
| PatchCore(FPFH) | 54.6 / 54.8 | 48.4 / 48.9 | 42.5 / 45.5 | 40.4 / 73.7 | 47.2 / 47.8 | 38.8 / 48.8 | 49.4 / 61.3 | 75.3 / 73.3 | 76.6 / 76.8 | 45.5 / 65.5 | 42.3 / 45.3 |
| PatchCore(PMAE) | 55.6 / 58.0 | 55.2 / 56.2 | 44.7 / 65.1 | 42.4 / 61.5 | 48.3 / 48.7 | 48.8 / 88.6 | 52.3 / 54.3 | 45.8 / 85.8 | 53.8 / 54.1 | 44.7 / 67.7 | 55.2 / 55.1 |
| CPMF | 55.5 / 55.5 | 58.9 / 54.2 | 46.2 / 51.5 | 52.0 / 52.0 | 61.0 / 61.1 | 50.9 / 50.9 | 68.5 / 78.3 | 35.9 / 45.8 | 69.7 / 65.7 | 45.1 / 45.8 | 34.5 / 48.6 |
| Reg3D-AD | 60.0 / 60.0 | 38.1 / 62.4 | 61.4 / 82.5 | 36.7 / 62.0 | 59.2 / 59.9 | 41.4 / 59.9 | 68.8 / 68.8 | 67.6 / 58.9 | 64.1 / 74.1 | 53.3 / 54.8 | 70.2 / 60.2 |
| ISMP | / 68.3 | / 62.2 | / 84.4 | / 72.2 | / 82.3 | / 66.1 | / 68.7 | / 52.2 | / 55.2 | / 66.1 | / 84.3 |
| Ours | 81.7 / 92.5 | 98.6 / 90.6 | 87.5 / 89.1 | 87.6 / 95.6 | 100. / 98.2 | 100. / 97.0 | 57.7 / 63.2 | 94.8 / 91.8 | 80.4 / 86.9 | 99.6 / 98.0 | 60.5 / 70.5 |

| Method | vase2 | vase3 | vase4 | vase5 | vase7 | vase8 | vase9 | Average |
|---|---|---|---|---|---|---|---|---|
| IMRNet | 61.4 / 61.4 | 70.0 / 40.1 | 52.4 / 52.4 | 67.6 / 68.2 | 63.5 / 59.3 | 63.0 / 63.5 | 59.4 / 69.1 | 66.1 / 65.0 |
| R3D-AD | 75.2 / | 74.2 / | 63.0 / | 75.7 / | 77.1 / | 72.1 / | 71.8 / | 74.9 / |
| PO3AD | 95.2 / 97.8 | 82.1 / 88.4 | 67.5 / 90.2 | 85.2 / 93.7 | 96.6 / 98.2 | 73.9 / 95.0 | 83.0 / 95.2 | 83.9 / 89.8 |
| BTF(RAW) | 41.0 / 40.3 | 71.7 / 60.2 | 42.5 / 61.3 | 58.5 / 58.5 | 44.8 / 57.8 | 42.4 / 55.0 | 56.4 / 56.4 | 49.3 / 55.0 |
| BTF(FPFH) | 54.6 / 64.6 | 69.9 / 69.9 | 51.0 / 71.0 | 40.9 / 42.9 | 51.8 / 54.0 | 66.8 / 66.2 | 26.8 / 56.8 | 52.8 / 62.8 |
| M3DM | 73.7 / 73.7 | 43.9 / 65.8 | 47.6 / 65.5 | 31.7 / 64.2 | 65.7 / 51.7 | 66.3 / 66.3 | 66.0 / 66.3 | 55.2 / 61.6 |
| PatchCore(FPFH) | 72.1 / 72.1 | 44.9 / 43.0 | 50.6 / 50.5 | 41.7 / 44.7 | 69.3 / 69.3 | 66.2 / 57.5 | 66.0 / 66.3 | 56.8 / 58.0 |
| PatchCore(PMAE) | 74.1 / 74.2 | 46.0 / 46.5 | 51.6 / 52.3 | 57.9 / 57.2 | 65.0 / 65.1 | 66.3 / 36.4 | 62.9 / 42.3 | 56.2 / 57.7 |
| CPMF | 58.2 / 58.2 | 58.2 / 58.2 | 51.4 / 51.4 | 61.8 / 65.1 | 39.7 / 50.4 | 52.9 / 52.9 | 60.9 / 54.5 | 55.9 / 57.3 |
| Reg3D-AD | 60.5 / 40.5 | 65.0 / 51.1 | 50.0 / 75.5 | 52.0 / 62.4 | 46.2 / 88.1 | 62.0 / 81.1 | 59.4 / 69.4 | 57.2 / 66.8 |
| ISMP | / 73.3 | / 76.2 | / 54.5 | / 47.2 | / 70.1 | / 85.1 | / 61.5 | / 69.1 |
| Ours | 100. / 99.7 | 84.5 / 84.4 | 81.8 / 92.7 | 100. / 87.9 | 64.3 / 86.3 | 81.8 / 93.4 | 87.3 / 97.1 | 86.1 / 88.2 |

M3DM [48], PatchCore [15], CPMF [49], Reg3D-AD [1], and ISMP [7]. On Real3D-AD, we additionally compare with Group3D [6]. Performance metrics for all compared methods are obtained from their original publications or publicly available implementations.

## 4.2 Quantitative results

### 4.2.1 Results on Real3D-AD

Table 1 presents the comparison of object-level and point-level anomaly detection performance on Real3D-AD. According to the average scores, our method achieves the best results on both metrics, outperforming the second-best method by 1.5% in O-AUROC and 4.2% in P-AUROC. Notably, our method attains the highest P-AUROC score in 8 categories and ranks second in one category, demonstrating its strong localization capability and confirming that the features learned by our model are highly discriminative.

### 4.2.2 Results on Anomaly-ShapeNet

Table 2 summarizes the results of anomaly detection and localization on Anomaly-ShapeNet. Our method achieves the best performance on O-AUROC, outperforming the second-best by 2.2%, and ranks second on P-AUROC with only a 1.6% gap to the top. Notably, Our approach significantly outperforms all memory bank-based methods, including Reg3D-AD and ISMP, on both O-AUROC (by 28.9%) and P-AUROC (by 19.1%), and achieves the best performance across 30 object categories.

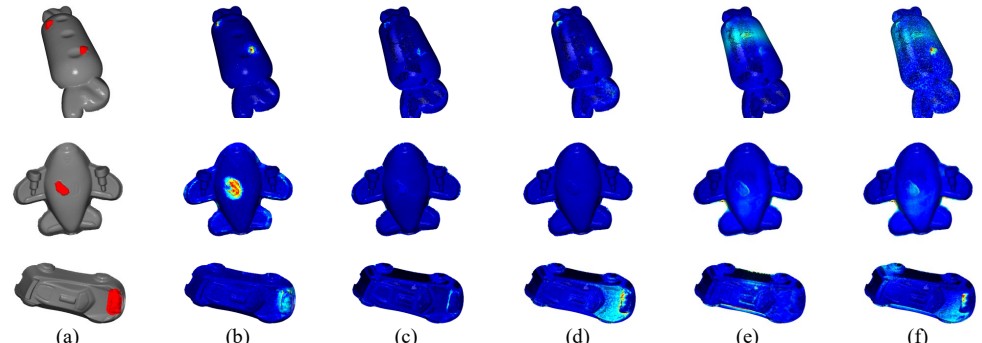

| (a) | (b) | (c) | (d) | (e) | (f) |

Figure 3: Visualization comparison of localization results on Real3D-AD. (a) Ground truth. (b) Ours. (c)&(e) PatchCore using FPFH and PointMAE, respectively. (d) ISMP. (f) Reg3D-AD.

These results show that our method learns rotation-invariant features with strong robustness and local discriminability, leading to more reliable 3D anomaly detection. The comparisons confirm its effectiveness and superiority, especially within memory bank-based frameworks.

### 4.3 Qualitative Results

As shown in Figure 3, our method generates sharper and more accurate anomaly maps on Real3D-AD, with detected anomalies closely aligning to the ground-truth defect regions. This leads to a significant improvement in P-AUROC compared to alternative approaches. Additional qualitative results are provided in the appendix.

### 4.4 Ablation Study

We conduct ablation studies on the registration strategy, training objective, memory bank feature composition, and performance under noisy point clouds to evaluate their respective effects. Additional ablation results are provided in the appendix.

**Evaluation on registration.** We evaluate the impact of different registration methods on anomaly detection performance using the Real3D-AD dataset. As shown in Table 3, two key insights emerge. First, when our registration method is applied, all downstream approaches achieve improved P-AUROC scores, demonstrating that it provides more accurate and stable alignment for anomaly localization compared to the commonly used FPFH+RANSAC. Second, even when using FPFH+RANSAC for registration, our method still outperforms competing approaches, indicating that our feature representations possess stronger rotation invariance and robustness properties essential for reliable 3D anomaly detection.

Table 3: Ablation study on registration strategy.

| Regis. | PatchCore (RAW) | PatchCore (PMAE) | Reg3D-AD | Ours |
|---|---|---|---|---|
| **F+R** | 65.3 | 64.2 | 70.5 | **72.6** |
| **Our** | 66.6 | 75.6 | 79.6 | **87.8** |

**Evaluation on training objective.** To assess the roles of $\mathcal{L}_{oc}$, $\mathcal{L}_f$, and $\mathcal{L}_p$, we performed a detailed ablation study on the Real3D-AD dataset. As shown in Table 4, $\mathcal{L}_{oc}$ is crucial for accurate registration, effectively aligning local patches even under partial overlap or sparsity, and its removal causes severe degradation. $\mathcal{L}_f$ enhances feature robustness by strengthening rotation invariance and local discriminability, thereby improving overall anomaly localization accuracy. $\mathcal{L}_p$ has limited effect on accuracy but ensures efficient and reliable point matching; without it, inference slows due to numerous spurious matches. Overall, $\mathcal{L}_{oc}$ and $\mathcal{L}_f$ enable accurate and stable detection, while $\mathcal{L}_p$ improves efficiency, jointly balancing performance and computational cost.

Table 4: Ablation study on training objective.

| loss | O-AUROC | P-AUROC |
|---|---|---|
| w/o $\mathcal{L}_{oc}$ | 58.1 | 68.1 |
| w/o $\mathcal{L}_p$ | 76.9 | 87.0 |
| w/o $\mathcal{L}_f$ | 73.4 | 86.0 |
| $\mathcal{L}_{oc} + \mathcal{L}_f + \mathcal{L}_f$ | **78.0** | **87.8** |

Table 5: Ablation study on memory bank feature composition.

| Real3D-AD | | | | Anomaly ShapeNet | | | |
|---|---|---|---|---|---|---|---|
| Coord. | Feat. | O-AUROC | P-AUROC | Coord. | Feat. | O-AUROC | P-AUROC |
| ✓ | | 70.3% | 66.2% | ✓ | | 80.5% | 67.2% |
| | ✓ | 76.4% | 83.7% | | ✓ | 82.4% | 85.0% |
| ✓ | ✓ | **78.0%** | **87.8%** | ✓ | ✓ | **86.1%** | **88.2%** |

Table 6: Ablation study on noisy point clouds.

| Setting 1 | | | Setting 2 | | |
|---|---|---|---|---|---|
| Metric | O-AUROC | P-AUROC | Noise | O-AUROC | P-AUROC |
| Clean | 78.0 | 87.8 | Clean | 78.0 | 87.8 |
| SD=0.001 | 76.7 | 87.5 | SD=0.001 | 76.9 | 86.5 |
| SD=0.003 | 77.7 | 86.4 | SD=0.003 | 63.0 | 69.1 |
| SD=0.005 | 76.8 | 83.1 | SD=0.005 | 56.6 | 56.5 |

**Evaluation on memory bank feature composition.** We evaluate the contribution of each memory bank feature component on Real3D-AD and Anomaly-ShapeNet. As shown in Table 5, coordinates (**Coord.**) and local features (**Feat.**) play complementary roles. **Coord.** provides global spatial cues, while **Feat.** captures fine-grained geometric details. Their combination achieves the best performance, showing strong synergy: **Coord.** locates anomalies globally, and **Feat.** enhances local discrimination. This is especially evident in geometric anomaly detection, where the combined use improves P-AUROC by 21.6% (Real3D-AD) and 21.0% (Anomaly-ShapeNet) over **Coord.**-only baselines.

**Evaluation on noisy point clouds.** We evaluate the robustness of our method to noisy point clouds on the Real3D-AD dataset under two settings. In **Setting 1**, Gaussian noise (SD = 0.005) is injected during training, with varying noise levels applied during testing. In **Setting 2**, noise is introduced only at test time. As shown in Tables 6, the model trained without noise augmentation (**Setting 2**) exhibits a clear performance drop at higher noise levels (SD $\geq$ 0.003), revealing its sensitivity to input perturbations. In contrast, noise-based augmentation during training (**Setting 1**) allows the model to maintain stable performance across different noise intensities.

## 5 Conclusion

In this paper, we propose Reg2Inv, a unified framework that leverages registration to learn rotation-invariant features for 3D anomaly detection. We identify key limitations in existing encoders, such as their poor preservation of fine-grained local geometry and lack of rotation invariance. To address these issues, we integrate registration into the feature learning process, enabling the extraction of structurally discriminative and transformation-robust representations. By jointly optimizing geometric alignment and multi-scale feature consistency within a unified training objective, our method produces features suitable for both registration and anomaly detection. Extensive experiments validate the effectiveness of Reg2Inv in achieving accurate registration and reliable anomaly detection.

**Limitation.** Despite promising results, our approach has several limitations. Training a model per class increases computational and storage costs. In less discriminative classes, the rotation-invariant feature extractor shows limited generalization. In addition, registration is not always accurate for anomaly detection. We plan to address these issues in future work to improve the method's efficiency and practicality.

**Acknowledgement.** This research is supported by the "Leading Talent" under Guangdong Special Support Program (2024TX08X048), China National Key R&D Program (2024YFB4709200), Key-Area Research and Development Program of Guangzhou City (No.2023B01J0022), Guangdong Provincial Natural Science Foundation for Outstanding Youth Team Project (No. 2024B1515040010), NSFC Key Project (No. U23A20391), Guangdong Natural Science Funds for Distinguished Young Scholars (Grant 2023B1515020097), the Singapore Ministry of Education AcRF Tier 1 Grant (Grant No.: MSS25C004), and the Lee Kong Chian Fellowships.

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

# Appendix

## A Method Details

### A.1 RIConv++

RIConv++ [39] is a rotation-invariant convolution method designed for deep learning on 3D point clouds. It proposes an Information-Rich Invariant Feature (IRIF) formulation that not only captures the relationship between the central point and its neighbors but also encodes the internal geometric relations among the neighboring points, thereby enhancing feature discriminability.

### A.2 Training objective details

$\mathcal{L}_f$ **for local feature alignment and** $\mathcal{L}_p$ **for point matching.** For each ground-truth patch match $(\mathcal{G}_{x_i}^{\mathcal{P}}, \mathcal{G}_{y_i}^{\mathcal{Q}}) \in \hat{\mathcal{M}}^{\text{gt}}$, two parallel optimal transport layers [42] are used to extract local feature assignment matrix and point assignment matrix, respectively. Specifically, we first compute cost matrices:

$$\mathcal{C}''_i = \mathbf{F}''_{x_i}^{\tilde{\mathcal{P}}} (\mathbf{F}''_{y_i}^{\tilde{\mathcal{Q}}})^T / d'',$$
$$\mathcal{C}'_i = \mathbf{F}'_{x_i}^{\tilde{\mathcal{P}}} (\mathbf{F}'_{y_i}^{\tilde{\mathcal{Q}}})^T / d', \tag{13}$$

where $\mathcal{C}''_i$ and $\mathcal{C}'_i \in \mathbb{R}^{n_i \times m_i}, n_i = |\mathcal{G}_{x_i}^{\mathcal{P}}|, m_i = |\mathcal{G}_{y_i}^{\mathcal{Q}}|$. The cost matrices $\mathcal{C}''_i$ are then augmented to $\mathcal{C}''^*_i$ by appending a new row and a new column, filled with a learnable dustbin parameter $\alpha$. We then utilize the Sinkhorn [43] algorithm on $\mathcal{C}''^*_i$ to compute a soft local feature assignment matrix $Z''^*_i$. The point assignment matrix $Z'^*_i$ is computed similarly. To prioritize local feature alignment and point matching, we apply separate **negative log-likelihood losses** [42] to the assignment matrices $Z''^*_i$ and $Z'^*_i$. The set of matched points is $\tilde{\mathcal{M}}^{\text{gt}}_i$. The sets of unmatched points in the two patches are denoted as $\mathcal{I}_i$ and $\mathcal{J}_i$. The individual local feature aligning loss $\mathcal{L}_{f,i}$ and point matching loss $\mathcal{L}_{p,i}$ are computed as:

$$\mathcal{L}_{f,i} = -\sum_{(x,y)\in\tilde{\mathcal{M}}^{\text{gt}}_i} \log z''^*_{i,x,y} - \sum_{x\in\mathcal{I}_i} \log z''^*_{i,x,m_i+1} - \sum_{y\in\mathcal{J}_i} \log z''^*_{i,n_i+1,y}, \tag{14}$$

$$\mathcal{L}_{p,i} = -\sum_{(x,y)\in\tilde{\mathcal{M}}^{\text{gt}}_i} \log z'^*_{i,x,y} - \sum_{x\in\mathcal{I}_i} \log z'^*_{i,x,m_i+1} - \sum_{y\in\mathcal{J}_i} \log z'^*_{i,n_i+1,y}, \tag{15}$$

The final losses are computed by averaging the individual loss: $\mathcal{L}_f = \frac{1}{N_g} \sum_{i=1}^{N_g} \mathcal{L}_{f,i}$, $\mathcal{L}_p = \frac{1}{N_g} \sum_{i=1}^{N_g} \mathcal{L}_{p,i}$.

$\mathcal{L}_{oc}$ **for patch matching.** To prioritize high-overlap matches for point cloud registration, we compute the **overlap-aware circle loss** [37, 44] on $\mathcal{P}$,

$$\mathcal{L}_{oc}^{\mathcal{P}} = \frac{1}{|\mathcal{A}|} \sum_{\mathcal{G}_i^{\mathcal{P}}\in\mathcal{A}} \log \Big[1 + \sum_{\mathcal{G}_j^{\mathcal{Q}}\in\varepsilon_p^i} e^{\lambda_i^j \beta_p^{i,j}(d_i^j-\Delta_p)} \cdot \sum_{\mathcal{G}_k^{\mathcal{Q}}\in\varepsilon_n^i} e^{\beta_n^{i,k}(\Delta_n-d_i^k)}\Big]. \tag{16}$$

Where $\mathcal{A}$ is the set of anchor patches in $\mathcal{P}$ that have at least one patch pair in $\hat{\mathcal{M}}^{\text{gt}}$. For each anchor patch $\mathcal{G}_i^{\mathcal{P}} \in \mathcal{A}$, $\varepsilon_p^i$ and $\varepsilon_n^i$ denote the sets of its positive and negative patches in $\mathcal{Q}$. $d_i^j = \|\mathbf{F}_i^{\hat{\mathcal{P}}} - \mathbf{F}_j^{\hat{\mathcal{Q}}}\|_2$ is the feature distance, and $\lambda_i^j = (o_i^j)^{\frac{1}{2}}$ and $o_i^j$ represents the overlap ratio between $\mathcal{G}_i^{\mathcal{P}}$ and $\mathcal{G}_j^{\mathcal{Q}}$. $\beta_p^{i,j} = \gamma(d_i^j - \Delta_p)$ and $\beta_n^{i,k} = \gamma(\Delta_n - d_i^k)$ are the positive and negative weights, respectively. We set the margins $\Delta_p = 0.1$ and $\Delta_n = 1.4$. The same applies to the loss $\mathcal{L}_{oc}^{\mathcal{Q}}$ in $\mathcal{Q}$ and the overall loss of the circle based on overlap is $\mathcal{L}_{oc} = (\mathcal{L}_{oc}^{\mathcal{P}} + \mathcal{L}_{oc}^{\mathcal{Q}})/2$.

The loss function $\mathcal{L} = \mathcal{L}_f + \mathcal{L}_p + \mathcal{L}_{oc}$ is composed of a feature aligning loss $\mathcal{L}_f$ for local feature aligning, a point matching loss $\mathcal{L}_p$ for point matching, and an overlap-aware circle loss $\mathcal{L}_{oc}$ for patch matching.

Table 7: Comparison of inference speed.

| Metric | RegAD | M3DM | ISMP | Ours |
|--------|-------|------|------|------|
| **AITPS** | 7.71 | 6.43 | 4.37 | **2.53** |

## A.3  Normalization parameters compute

For each train sample $\mathcal{P} \in \mathcal{D}_{\text{train}}$, we derive local features $\mathbf{F}''^{\tilde{\mathcal{P}}} \in \mathbb{R}^{|\tilde{\mathcal{P}}| \times d''}$ and the corresponding alignment coordinates $\tilde{\mathcal{P}}^{align} \in \mathbb{R}^{|\tilde{\mathcal{P}}| \times 3}$. These are then aggregated across all training samples in $\mathcal{D}_{\text{train}}$ to form collections of feature vectors $\mathbf{F}_f$ and coordinate vectors $\mathbf{F}_c$, from which we compute the normalization parameters $\gamma_f$ and $\gamma_c$, respectively:

$$\begin{cases} \gamma_f = \max_{i \in \{1,...,N\}} \|\mathbf{F}_f^{(i)}\|_2, \mathbf{F}_f = \bigcup_{i=1}^{M} \mathbf{F}''^{\tilde{\mathcal{P}}}_i, \\ \gamma_c = \max_{i \in \{1,...,N\}} \|\mathbf{F}_c^{(i)}\|_2, \mathbf{F}_c = \bigcup_{i=1}^{M} \tilde{\mathcal{P}}^{align}_i, \end{cases} \tag{17}$$

where $M$ denotes the total number of samples in $\mathcal{D}_{\text{train}}$ and $N$ represents the dimensionality of the feature vectors.

## A.4  AUROC

We evaluate anomaly detection performance at both the object and point levels using the Area Under the Receiver Operating Characteristic Curve (AUROC). The AUROC quantifies the overall discriminative capability of a model. Given prediction scores $s(x)$ for samples $x$, the true positive rate (TPR) and false positive rate (FPR) at a decision threshold $\tau$ are defined as

$$\text{TPR}(\tau) = \frac{\text{TP}(\tau)}{\text{TP}(\tau) + \text{FN}(\tau)}, \quad \text{FPR}(\tau) = \frac{\text{FP}(\tau)}{\text{FP}(\tau) + \text{TN}(\tau)}. \tag{18}$$

The Receiver Operating Characteristic (ROC) curve plots $\text{TPR}(\tau)$ against $\text{FPR}(\tau)$ over all thresholds $\tau \in [0, 1]$. The AUROC is defined as the area under this curve:

$$\text{AUROC} = \int_0^1 \text{TPR}(\text{FPR}) \, d(\text{FPR}). \tag{19}$$

In discrete form, the AUROC can be approximated as

$$\text{AUROC} = \frac{1}{N_+ N_-} \sum_{i=1}^{N_+} \sum_{j=1}^{N_-} \mathbb{I}\big(s(x_i^+) > s(x_j^-)\big), \tag{20}$$

where $N_+$ and $N_-$ denote the number of positive and negative samples, and $\mathbb{I}(\cdot)$ is the indicator function.

## A.5  Training Efficiency

Our method learns features via a registration task that enforces geometric alignment and multi-scale consistency between source and target point clouds. As it does not rely on category-specific supervision, it is inherently category-agnostic and can be trained jointly across the entire dataset. Under our experimental setup (RTX 3090 GPU, batch size 1), full training takes approximately 27 hours on Anomaly-ShapeNet (40 categories) and about 34 hours on Real3D-AD (12 categories). In contrast, methods like R3D-AD and PO3D-AD require  1 hour and  7 hours per category, respectively. While our single training run is longer, it eliminates the need for category-wise training, resulting in significantly greater overall efficiency.

## A.6  Inference Efficiency

To assess runtime efficiency, we report the **average inference time per sample** (AITPS, measured in seconds). Using an RTX 3090 GPU with a batch size of 1, our method achieves an average processing time of 2.53 seconds per Real3D-AD sample, which is faster than other memory bank-based approaches, as shown in Tables 7.

# B   More Ablation Studies

To better understand the role of different design choices in our framework, we perform ablation studies on four critical components: (i) the computation method for object-level anomaly scoring; (ii) the neighborhood size in RIConv++; (iii) normalization and filtering of features during inference; (iv) using alternative extracted features for memory bank construction. These ablations provide insights into how each component contributes to overall performance and robustness. Additionally, to evaluate the robustness of our method, we conduct an experiment with varying rotation angles of point clouds, demonstrating its stability under different geometric transformations.

## B.1   Evaluation on the calculation method of object-level anomaly score.

We evaluate the impact of different strategies for computing the object-level anomaly score(%) on the Anomaly-ShapeNet dataset. We compare two standard aggregation methods: **mean**, defined as $S = mean(\{s_i\})$, and **max**, defined as $S = max(\{s_i\})$. Our proposed method, denoted as **ours**, is formulated

Table 8: Ablation study on calculation method.

| Method | mean | max | ours |
|--------|------|-----|------|
| O-AUROC | 81.6 | 82.8 | 86.1 |

as $S = \max(\{s_i\} * f_n)$, where $s_i$ is the point-level anomaly score, $f_n$ is a mean filter of size $n$, and $*$ is the point-wise convolution operator. As shown in Table 8, our method achieves the highest O-AUROC score among all aggregation strategies. Unlike conventional mean or max operations, our approach first applies local smoothing followed by a max operation, which effectively highlights true anomalous regions while reducing the impact of noise and unreliable point-level predictions.

## B.2   Evaluation on neighborhood size in RIConv++.

Figure 4 shows object-level and point-level AU-ROC results with varying neighborhood sizes in RIConv++, evaluated on the Real3D-AD dataset. Selecting an appropriate neighborhood size is critical: an excessively large size may dilute geometric patterns by aggregating diverse points, weakening local feature discrimination; conversely, an overly small size may limit contextual coverage, reducing the expressiveness of local structures. Despite these effects, our method remains relatively robust to neighborhood size choices. As shown in Figure 4, detection and localization performance peak at patch numbers 32 and 64, respectively. We set the patch num-

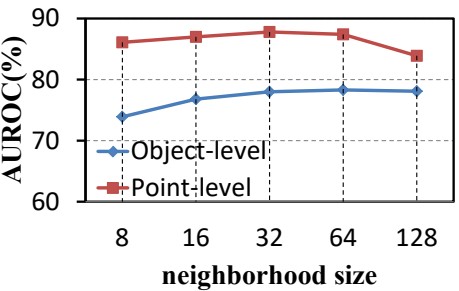

Figure 4: Ablation study on neighborhood sizes.

ber to 32 in our implementation to prioritize detection performance, with only a minor trade-off in localization accuracy.

## B.3   Evaluation on Normalization & Filter in inference phase.

We evaluate the impact of feature normalization and filtering strategies during the inference phase on both object-level and point-level anomaly detection performance(%) using the Real3D-AD dataset. As shown in Table 9, normalization emerges as the key component driving performance improvement. Feature normalization alone improves O-AUROC by 10.5% and P-AUROC by 21.6%. In contrast, filtering plays a more auxiliary role. Applying filtering alone results in only marginal gains, but when combined with normalization, it significantly enhances performance, achieving an O-AUROC improvement of 17.9% and a P-AUROC improvement of 22.5%. These results highlight the importance of proper feature conditioning during inference and suggest a synergistic effect between normalization and filtering in improving model reliability and sensitivity.

Table 9: Ablation study on Normalization & Filter.

| Normalization | Filter | O-AUROC | P-AUROC |
|:---:|:---:|:---:|:---:|
| | | 60.1 | 65.3 |
| | ✓ | 62.9 | 65.7 |
| ✓ | | 70.6 | 86.9 |
| ✓ | ✓ | 78.0 | 87.8 |

## B.4 Evaluation on alternative feature sources for memory bank construction

We conducted an ablation study on alternative feature representations for memory bank construction using the Real3D-AD dataset. As shown in Table 10, KPConv-FPN features $\mathbf{F}'$ are suboptimal due to limited receptive fields and weaker semantics.

Table 10: Ablation study on the memory bank construction.

| Memory Bank Construction | O-AUROC | P-AUROC |
|:---:|:---:|:---:|
| $\mathbf{F}'$ | 51.1 | 43.8 |
| $\mathcal{P}^{align} + \mathbf{F}'$ | 54.0 | 56.1 |
| $\mathbf{F}''$ | 76.4 | 83.7 |
| $\mathcal{P}^{align} + \mathbf{F}''$ | **78.0** | **87.8** |

## B.5 Evaluation on rotated point clouds

We conducted a rotation experiment on the Real3D-AD dataset to directly demonstrate the enhanced rotation invariance and robustness of our learned features. At test time, each point cloud is randomly rotated around one of the X, Y, or Z axes by 90°, 180°, or 270°. We then evaluate the performance of the model using only the features learned in these rotations. As shown in Table 11, the model achieves consistent performance at all angles, confirming that the learned features possess strong rotation invariance and generalizability.

Table 11: Results under varying levels of rotation.

| Rotation Angle | O-AUROC | P-AUROC |
|:---:|:---:|:---:|
| 0° | 76.4 | 83.7 |
| 90° | 76.3 | 83.9 |
| 180° | 76.2 | 83.6 |
| 270° | 76.6 | 84.3 |

# C More Qualitative Results

We present additional qualitative results to further demonstrate the anomaly localization capability of our method. Figure 5 shows the results on the Real3D-AD dataset, while Figures 6 and 7 display the results on the Anomaly-ShapeNet dataset. These visualizations highlight that our method not only accurately identifies anomalous regions but also produces spatially coherent anomaly scores, effectively distinguishing between normal and defective structures in 3D point clouds.

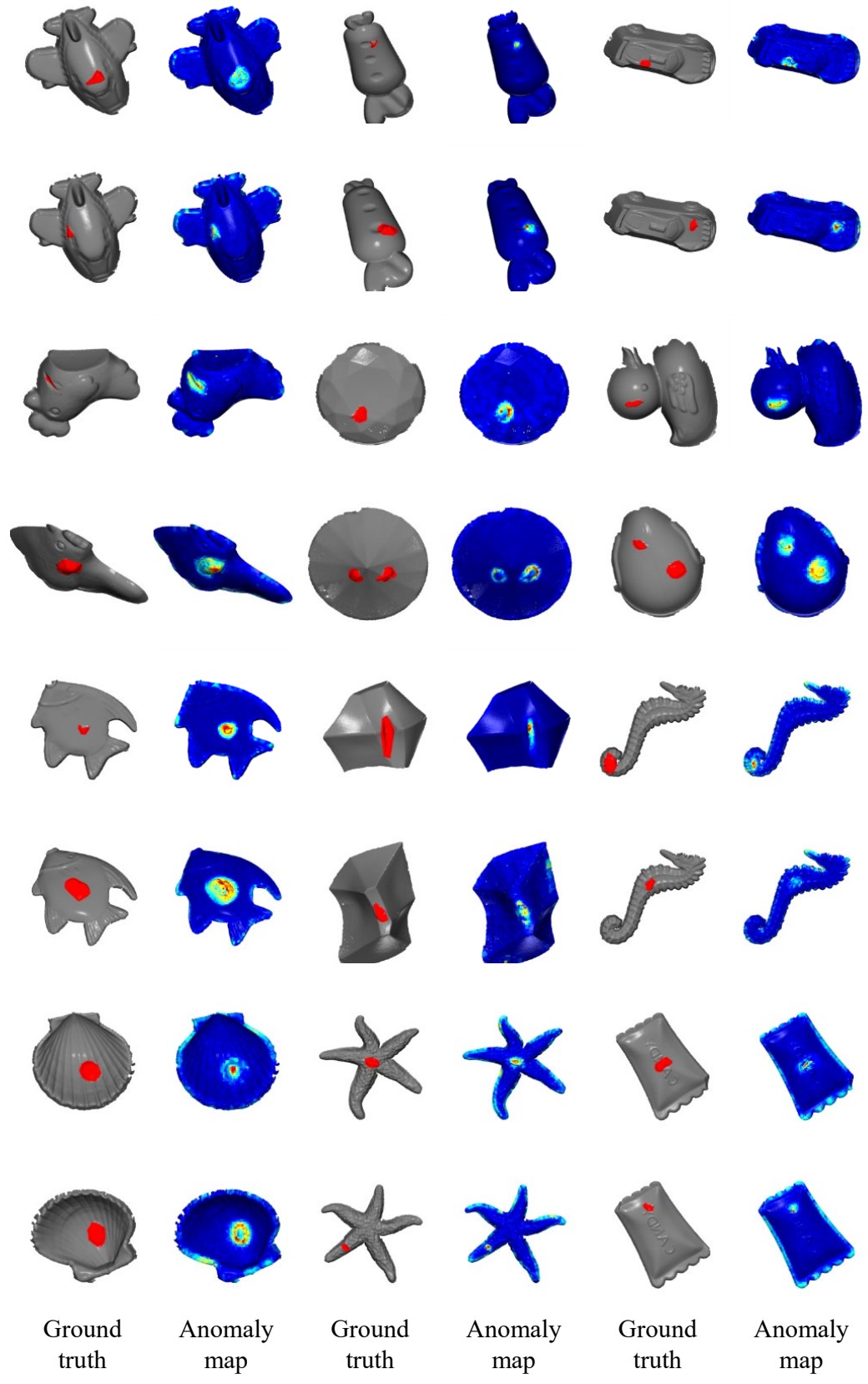

Ground truth    Anomaly map    Ground truth    Anomaly map    Ground truth    Anomaly map

Figure 5: More qualitative results of localization on the Real3D-AD dataset.

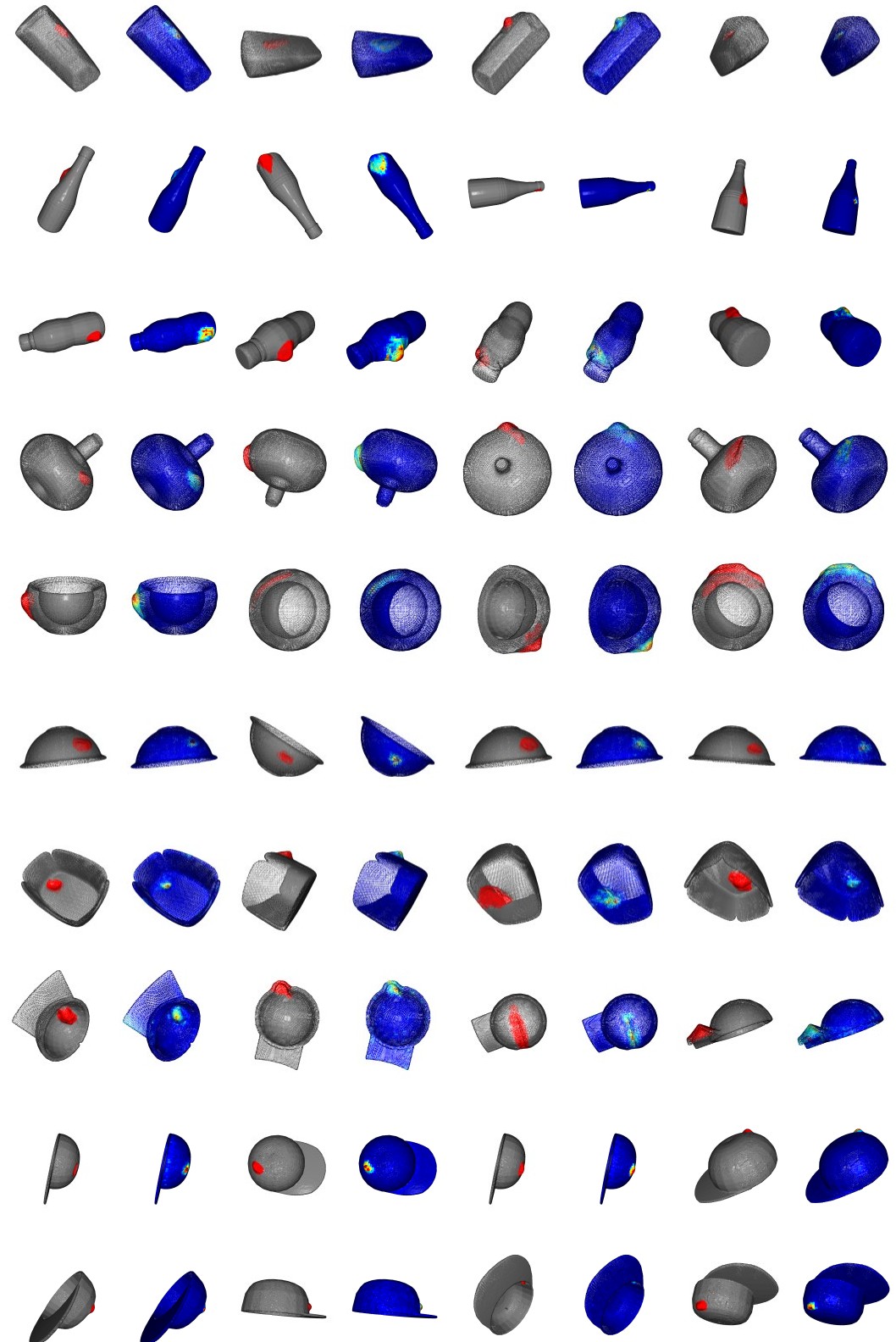

Ground truth | Anomaly map | Ground truth | Anomaly map | Ground truth | Anomaly map | Ground truth | Anomaly map

Figure 6: More qualitative results of localization on the Anomaly-Shapenet dataset.

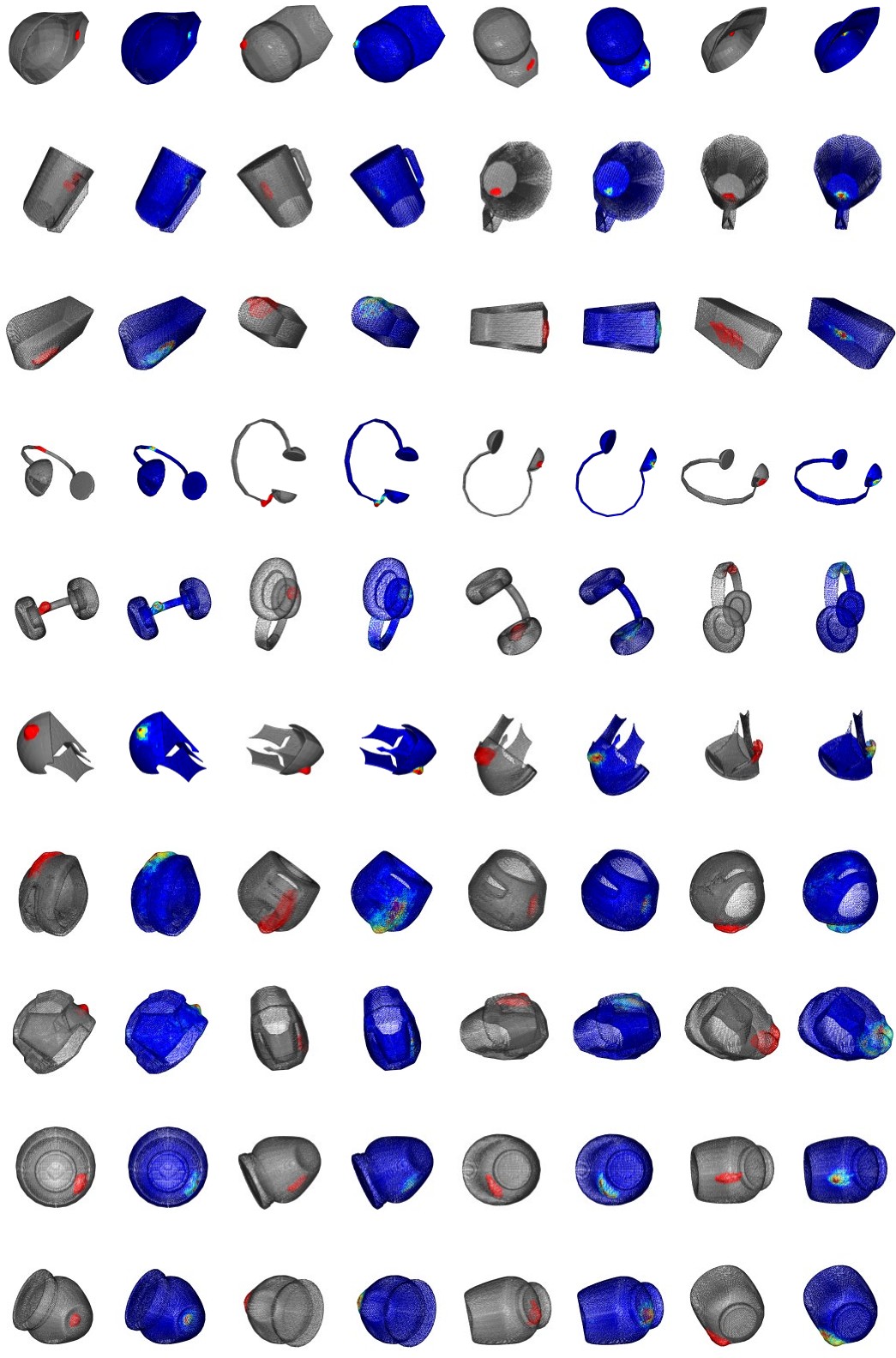

Ground truth    Anomaly map    Ground truth    Anomaly map    Ground truth    Anomaly map    Ground truth    Anomaly map

Figure 7: More qualitative results of localization on the Anomaly-Shapenet dataset.

