# OpenReview forum: "Registration is a Powerful Rotation-Invariance Learner for 3D Anomaly Detection"
_NeurIPS.cc/2025/Conference — NeurIPS 2025 poster_

### Official Review · Reviewer_Zbiu · 2025-06-01

**Clarity:** 2
**Significance:** 3
**Originality:** 3
**Rating:** 4
**Confidence:** 4

**Summary:**

This paper introduces a novel framework called Reg2Inv that integrates point cloud registration with feature learning to achieve rotation-invariant and locally discriminative representations for 3D anomaly detection. The key idea is to leverage the alignment process of registration to guide feature extraction, ensuring that the learned features are robust to rotations and effective for identifying structural defects. Experimental results demonstrate that RegInv outperforms existing methods on the Anomaly-ShapeNet and Real3D-AD datasets.

**Questions:**

According to weaknesses, my questions are:
1. The introduction and related work sections could provide a more thorough discussion of the limitations of existing methods and how Reg2Inv addresses these limitations. This would help readers understand the significance of the contributions more clearly.
2. The evaluation metrics lack detailed explanations. It is recommended to supplement with calculation formulas and the set thresholds.
3. The paper compares its method with several state-of-the-art approaches but does not provide a detailed analysis of why Reg2Inv outperforms these methods. A deeper comparative analysis, including qualitative results and discussions on specific cases where Reg2Inv excels, would enhance the validation of the claims.
4. The authors indirectly verify that the learned features possess stronger rotation invariance and robustness properties through the ablation study on registration. However, this is not intuitive. It is suggested that the authors apply different levels of rotation to the two datasets and test the performance changes of different methods.

**Ethical Concerns:**

["NO or VERY MINOR ethics concerns only"]

**Final Justification:**

My concerns regarding the experimental results have been addressed. I plan to raise my score to borderline accept.

**Limitations:**

Yes

**Quality:**

2

**Strengths And Weaknesses:**

Strengths:
1. The writing quality of the paper is excellent. The authors clearly articulate the problem, review related work, and present their methodology logically and coherently.
2. The idea of this paper is interesting. Using registration to learn rotation invariance for anomaly detection is intuitive and in line with the direction of research development.

Weaknesses:
1. The visualizations in Fig. 1 are not aligned.
2. The evaluation metrics are not described clearly.
3. Experiments lack detailed analysis of why Reg2Inv outperforms these methods.

---

> ### Author Rebuttal · Authors · 2025-07-31
>
> #### **Point-by-Point Response to Reviewer Zbiu**
> We sincerely appreciate your insightful comments. We believe some concerns may stem from misunderstandings, which we clarify below.
>
> > **Question-1:** *" The visualizations in Fig. 1 are not aligned. In addition, the introduction and related work sections could provide a more thorough discussion of the limitations of existing methods and how Reg2Inv addresses these limitations. "*
> >
> > **Response:** We thank the reviewer for the comment. However, it appears there may have been a misunderstanding regarding the role of **Figure 1**, as well as the depth of discussion provided in the manuscript.
> >
> > **Figure 1, along with the second through fourth paragraphs of the Introduction**, was specifically designed to articulate the motivation behind our work and to highlight the limitations of existing methods. In particular, this section discusses two critical challenges:
> >
> > 1. **Unreliable Registration:** Traditional methods fail under large pose variations (Fig. 1(a)), preventing accurate spatial alignment.
> > 2. **Weak Feature Representation:** Methods like PointMAE lack local discriminability and rotation invariance (Fig. 1(b)), leading to inconsistent features and degraded anomaly detection.
> >
> > Our method, **Reg2Inv**, is explicitly proposed to overcome these challenges by jointly optimizing registration and feature learning to produce robust, locally discriminative features (Fig. 1(d)), improving both alignment (Fig. 1(c)) and detection performance.
> >
> > We welcome constructive suggestions to improve the clarity and precision of our writing. However, a single, high-level comment without concrete guidance does not provide actionable feedback for revision and is unlikely to benefit the submission or the broader research community. We are fully committed to improving the manuscript and would greatly appreciate more specific comments on how to strengthen the discussion or presentation.
>
>
> > **"Question-2:"** *" The evaluation metrics lack detailed explanations. It is recommended to supplement with calculation formulas and the set thresholds. "*
> >
> > **Response:** Thank you for the suggestion. The evaluation metrics we use are standard and widely adopted in the literature, and, in line with common practice, we did not include the calculation formulas in the main paper to preserve space and maintain clarity. However, as suggested, we will provide detailed formulas and threshold settings in the supplementary material.
>
> > **"Question-3:"** *" The paper compares its method with several state-of-the-art approaches but does not provide a detailed analysis of why Reg2Inv outperforms these methods. A deeper comparative analysis, including qualitative results and discussions on specific cases where Reg2Inv excels, would enhance the validation of the claims. "*
> >
> > **Response:** We respectfully disagree with this comment. Due to space constraints, we have made our best effort to provide a detailed comparative analysis throughout the main paper and supplementary material. In addition to the experimental section, we have integrated comparative insights into **Figure 1**, **Figure 3**, and conducted **extensive experiments in the supplementary material**, all of which collectively explain why Reg2Inv outperforms existing methods.
> >
> > Specifically, Reg2Inv excels by effectively capturing discrepancies in both coordinates and features:
> >
> > 1. **Improved Registration Accuracy:** As shown in **Fig. 1(a) and (c)**, Reg2Inv achieves better alignment, refining spatial correspondence and reducing false geometric deviations. This allows coordinate differences to more faithfully reflect structural anomalies.
> >
> > 2. **Enhanced Feature Discriminability:** As demonstrated in **Fig. 3**, Reg2Inv produces locally discriminative features that remain consistent in normal regions and highlight anomalies more clearly. Competing methods tend to exhibit missed detections or false positives due to weaker local feature modeling.
> >
> > 3. **Rotation Invariance without Memory Bank:** **Table 3** in the main text shows that Reg2Inv outperforms memory bank-based methods using only local features, confirming its robustness to rotation and strong feature consistency.
> >
> > 4. **Comprehensive Ablation Studies:** The supplementary material includes **extensive ablation studies** that evaluate the individual contributions of each component in our framework, further validating the effectiveness of our design choices.
> >
> > We appreciate the suggestion and will further strengthen this analysis in the revised experimental section. However, we believe the current version already provides a thorough and well-supported explanation of Reg2Inv’s advantages.
>
>
> > **"Question-4:"** *" It is not intuitive to verify that the learned features possess stronger rotation invariance and robustness properties through the ablation study on registration. It is suggested that the authors apply different levels of rotation to the two datasets and test the performance changes of different methods. "*
> >
> > **Response:** We acknowledge that the ablation study on registration does not directly demonstrate the enhanced rotation invariance and robustness of our learned features. To address this, we conducted a rotation experiment on the Real3D-AD dataset. At test time, each point cloud was randomly rotated around one of the X, Y, or Z axes by 90°, 180°, or 270°. We then evaluated the model’s performance using only the learned features under these rotations. The results in **Table 1** show consistent performance, confirming that the learned features possess strong rotation invariance and generalization ability.
> >
> > **Table 1:** Results under varying levels of rotation.
> > | rotation angle | O-AUROC | P-AUROC |
> > | :---: | :---: | :---: |
> > | 0° | 76.4 | 83.7 |
> > | 90° | 76.3 | 83.9 |
> > | 180° | 76.2 | 83.6 |
> > | 270° | 76.6 | 84.3 |
>
>
>
> **To Reviewer Zbiu:** We sincerely thank you for your time and thoughtful comments. However, it seems that some concerns may stem from a partial misunderstanding of our work. In particular, the key issues raised regarding the level of comparative analysis and discussion appear to overlook the efforts we have made throughout the manuscript, including detailed explanations in the main text, visual illustrations in Figures 1 and 3, and extensive experiments and ablation studies in the supplementary material.
>
> We fully respect the importance of constructive critique and welcome suggestions that help improve the quality of the work. However, some of the comments are quite general and would benefit from more specific guidance to better serve both the revision process and the research community.
>
> We have carefully addressed all concerns in our rebuttal and sincerely hope that, with these clarifications, the contributions of our work can be reconsidered in a fair and balanced manner.

---

> ### Author Response · Authors · 2025-08-06
> **Follow-Upon Rebuttal Response**
>
> Dear Reviewer Zbiu,
>
> Thank you for your thoughtful and constructive feedback. We have provided detailed responses to your concerns above. We notice that you have not yet engaged in further discussion with us. If there are any remaining or unclear issues, we would greatly appreciate the opportunity to clarify or further improve our work. Your feedback is highly valuable, and we are eager to address any remaining points that could help strengthen the paper.
>
> Sincerely,
>
> The Authors

---

### Official Review · Reviewer_iakd · 2025-06-26

**Clarity:** 2
**Significance:** 3
**Originality:** 3
**Rating:** 3
**Confidence:** 4

**Summary:**

This paper proposes a 3D industrial anomaly detection method called Reg2Inv. This approach achieves better registration effects and ensures more balanced coverage of discriminative features in the memory bank. The proposed method was evaluated on the Real3D-AD and Anomaly-ShapeNet datasets, and was compared against both recently proposed and classic approaches, demonstrating certain advantages.Additionally, ablation studies were conducted on the key modules to evaluate their individual contributions.

**Questions:**

About the generalizability of registration-based methods. Is the proposed registration based method still maintains effectiveness in 2D scenarios (e.g., Real-IAD dataset) or multimodal settings (e.g., MVTec3D-AD) ?

**Ethical Concerns:**

["NO or VERY MINOR ethics concerns only"]

**Limitations:**

yes

**Quality:**

2

**Strengths And Weaknesses:**

Strength:

1.Overall, 3D anomaly detection holds considerable research value.

2. The motivation behind the proposed method is both reasonable and highly interesting. Registration indeed plays a critical role in anomaly detection, and the consideration of disriminative features coverage across the entire object in the memory bank is technically sound. Notably, this is the first work I've encountered that addresses this specific aspect, demonstrating clear innovativeness.


Weakness

1.The overall clarity and readability of the method presentation need improvement, particularly in Figure 2's framework illustration. The Rotation-Invariant Feature Extractor is an independent module that doesn't necessarily need to be included in Figure 2, as this unnecessarily increases the diagram's complexity. Additionally, the excessive use of different dashed line types makes the figure difficult to interpret, especially in printed versions.

2.In the comparative experiments, the proposed method demonstrates certain advantages over other methods, though the improvements are not statistically significant.

3. The presentation of experimental tables has significant issues. Table 1 and Table 2 compare too many methods, resulting in fragmented content that compromises clarity. The Experiment Setting section should clearly specify which representative methods are truly worthy of comparison.
Regarding Table 2, the Anomaly-Net results include excessive object categories that unnecessarily complicate the main text. The paper should focus on reporting average metrics in the main content while moving detailed breakdowns to supplementary materials. Additionally, the experimental analysis needs strengthening to better contextualize these results.
Table 3 suffers from an awkward format, combining two internally unrelated tables into a single presentation. This layout choice hinders clear interpretation of the results.

---

> ### Author Rebuttal · Authors · 2025-07-31
>
> #### **Point-by-Point Response to Reviewer iakd**
> We sincerely appreciate your insightful comments. We believe some concerns may stem from misunderstandings, which we clarify below.
>
> > **Question-1:** *" The overall clarity and readability of the method presentation need improvement, particularly in Figure 2's framework illustration. "*
> >
> > **Response:** Our intention was to align the architectural diagram with the methodology by illustrating the training/inference procedures, as well as the Rotation-Invariant Feature Extractor. We acknowledge the resulting complexity and will simplify Figure 2 by removing unnecessary lines, standardizing dashed styles, and eliminating redundant labels.
>
> > **Question-2:** *" In comparative experiments, the proposed method shows some advantages over others, though the improvements are not statistically significant. "*
> >
> > **Response:** We respectfully disagree with the statement that the improvements are not statistically significant. As shown in **Tables 1 and 2**, our method achieves clear and consistent performance gains over existing approaches across both benchmarks:
> >
> > * On **Real3D-AD**, our method achieves an average **O-AUROC of 78.0%** and **P-AUROC of 87.8%**, outperforming the second-best with **76.5% / 83.6%**, a margin of **+1.5% and +4.2%**, respectively.
> > * On **Anomaly-ShapeNet**, our method achieves **86.1% / 88.2%**, compared to **83.9% / 89.8%** by PO3AD. These improvements in object-level AUROC reflect stronger detection ability on clean and complete shapes.
> >
> > We believe these gains are significant and reflect the robustness and effectiveness of our approach. Moreover, reconstruction-based methods like PO3AD rely on complete geometry and cannot operate on partial or occluded point clouds. As a result, they struggle to achieve satisfactory performance when computing point-level AUROC on Real3D-AD and are limited to classification only. In contrast, our method processes incomplete inputs and supports both classification and localization, offering broader applicability in real-world scenarios.
> >
> > In summary, the consistent performance advantage, along with greater flexibility, demonstrates the practical strength and generalization ability of our method.
>
>
> > **Question-3:** *" The presentation of experimental tables has significant issues in Tables 1-3, including overcrowding with too many methods, overly complex content, and suboptimal formatting. Additionally, the experimental analysis needs strengthening to better contextualize these results. "*
> >
> > **Response:** We thank the reviewer for pointing this out. While the design of Tables 1 and 2 follows common practices in prior work, we agree that the large number of object categories may make the tables appear visually dense. However, this level of detail is necessary for a fair and comprehensive comparison. To aid interpretation, we would like to emphasize that the **average scores at the end of each table provide the most intuitive summary of overall performance**, and these averages are the key indicators we highlight in the analysis.
> >
> > To further improve clarity, we will restructure Tables 1 and 2 by grouping methods by category and simplifying the layout of Table 2. Table 3 will be split into two separate tables for easier readability.
> >
> > In addition, we will expand the experimental analysis with more focused discussion to better contextualize the results and clearly highlight the strengths of our method.
>
>
> > **Question-4:** *" Is the proposed registration-based method still maintains effectiveness in 2D scenarios(e.g., Real-IAD dataset) or multimodal settings (e.g., MVTec3D-AD) ? "*
> >
> > **Response:** We thank the reviewer for raising this question, and we discuss it from two perspectives to provide a comprehensive response：
> >
> > 1. **Application to 2D Anomaly Detection:** 2D registration typically involves local, pixel-wise alignment, making it challenging to estimate a reliable global transformation between images captured from different viewpoints. Without consistent global alignment, holistic feature extraction becomes unreliable, and the benefits of 3D registration cannot be achieved. As such, our registration-based approach is not directly applicable to 2D anomaly detection.
> >
> > 3. **Application to Multimodal Anomaly Detection:** For multimodal anomaly detection on the MVTec3D-AD dataset, which provides an RGB image and a corresponding point cloud from a single viewpoint, anomaly detection can be performed on the point cloud using our registration-based method, with the results subsequently fused with those from the image modality. However, as MVTec3D-AD provides only partial point clouds, this prevents the use of a complete shape template, potentially leading to less stable registration.
>
>
> **To Reviewer iakd:** The primary concern raised appears to be that our performance improvements are not statistically significant. However, this may stem from overlooking the average performance metrics reported in each table, where our method consistently achieves clear and meaningful gains over all baselines. The other concerns, such as the complexity of Figure 2 and the formatting of the tables, are relatively minor and can be readily addressed during revision.
>
> Given that the core issues raised are either not substantiated by the data or are minor in nature, we believe the negative evaluation is not justified. We have addressed all points thoroughly in our rebuttal and sincerely request a reconsideration of the decision in light of the significant, consistent, and practically meaningful improvements demonstrated by our method.

---

> ### Author Response · Authors · 2025-08-06
> **Follow-Upon Rebuttal Response**
>
> Dear Reviewer iakd,
>
> Thank you for your thoughtful and constructive feedback. We have provided detailed responses to your concerns above. We notice that you have not yet acknowledged the rebuttal, submitted your final recommendation, and engaged in further discussion with us. If there are any remaining or unclear issues, we would greatly appreciate the opportunity to clarify or further improve our work. Your feedback is highly valuable, and we are eager to address any remaining points that could help strengthen the paper.
>
> Sincerely,
>
> The Authors

---

> ### Author Response · Authors · 2025-08-07
> **Gentle Reminder: Discussion, Acknowledgement, and Final Rating Pending**
>
> Dear Reviewer iakd,
>
> With the Reviewer Author Discussion phase nearing its deadline, we kindly noticed that the **Discussion, Mandatory Acknowledgement, and Final Rating** are still pending on your end. We would be sincerely grateful if you could take a moment to complete these final steps at your earliest convenience.
>
> Sincerely,
>
> The Authors

---

> ### Author Response · Authors · 2025-08-07
> **Reminder: Discussion, Acknowledgement, and Final Rating**
>
> Dear Reviewer iakd,
>
> As the Reviewer Author Discussion phase is coming to an end, we noticed that you **have not yet participated in the discussion, submitted the Mandatory Acknowledgement, or provided the Final Rating.**
>
> We would be sincerely grateful if you could kindly take a moment to complete these final steps at your earliest convenience.
>
> Thank you very much for your time and consideration.
>
> Sincerely,
>
> The Authors

---

> ### Comment · Area_Chair_L7NC · 2025-08-08
>
> Dear Reviewers **iakd** and **BTG1**,
>
> Could you please share any additional thoughts you have on the rebuttal and the comments from the other reviewers? Your involvement in this author-reviewer discussion would be greatly appreciated. Thank you very much for your time and effort.
>
> Best,
>
> AC

---

### Official Review · Reviewer_BTG1 · 2025-07-01

**Clarity:** 4
**Significance:** 4
**Originality:** 3
**Rating:** 5
**Confidence:** 3

**Summary:**

The authors propose Reg2Inv, an anomaly detection framework where the core insight is the connection between anomaly detection and point cloud registration. In the first stage, the model learns feature via a registration task, while in the inference stage, rotation-invariant features are extracted and used in a memory-bank-based anomaly detection approach. The proposed approach achieves excellent results on standard datasets.

**Questions:**

- In the ablation study, Table 3: How come the Coord works fairly well? It does not seem to contain that much information intuitively.
- What is the intuition behind using F’’ and the P^align coordinates as the features in the coreset? Why not other extracted features?

**Ethical Concerns:**

["NO or VERY MINOR ethics concerns only"]

**Final Justification:**

Most of the issues I've raised have been addressed in the rebuttal. I maintain my score.

**Limitations:**

The authors have adequately addressed the limitations.

**Paper Formatting Concerns:**

None.

**Quality:**

4

**Strengths And Weaknesses:**

Strengths:
- Network is trained for registration which then learns to extract features that perform well in memory-based anomaly detection models. The insight that performing registration produces features that are discriminative enough for anomaly detection is interesting.
- The proposed method is conceptually simple but interesting and is based on a novel insight.
- The paper is generally very well written and easy to follow.

Weaknesses:
- The main major weakness is the somewhat weak ablation study. No ablation experiments in the paper for Loss components, filtering, using other extracted features for memory bank computation etc.
- Lacking intuition for some design choices such as various loss components, filtering etc. Essentially things that are missing from the ablation study. I know these are present in the quite extensive supplementary material but it would still be useful for the reader to see more of these in the main paper.

---

> ### Author Rebuttal · Authors · 2025-07-31
>
> #### **Point-by-Point Response to Reviewer BTG1**
> We sincerely appreciate your insightful comments, which significantly improved our paper. In response, we provide our detailed replies to your questions as follows:
>
> > **Question-1:** *" The main major weakness is the somewhat weak ablation study. No ablation experiments in the paper for Loss components, filtering, using other extracted features for memory bank computation, etc. "*
> >
> > **Response:** We thank the reviewer for the valuable feedback on the ablation study. We would like to clarify that several ablations addressing key components of our method are included in the supplementary material. Specifically, Table 1 presents the ablation of the loss component $\mathcal{L}\_{f}$, highlighting its critical contribution to detection performance. Table 2 reports the impact of the filtering strategy, which serves an auxiliary function.
> >
> > Additionally, we conducted an ablation study on alternative feature representations for memory bank construction using the Real3D-AD dataset. As shown in **Table 1** below, KPConv-FPN features $\mathbf{F}'$ are suboptimal due to limited receptive fields and weaker semantics.
> >
> > **Table 1:** Ablation study on the use of extracted features for memory bank construction.
> > | Memory Bank Construction | O-AUROC | P-AUROC |
> > | :---: | :---: | :---: |
> > | $\mathbf{F}'$ | 51.1 | 43.8 |
> > | $\mathcal{P}^{align}$ + $\mathbf{F}'$ | 54.0 | 56.1 |
> > | $\mathbf{F}''$ | 76.4 | 83.7 |
> > | $\mathcal{P}^{align}$ + $\mathbf{F}''$ | 78.0 | 87.8 |
>
> > **Question-2:** *" Lacking intuition for some design choices such as various loss components, filtering, etc. Essentially, things that are missing from the ablation study. I know these are present in the quite extensive supplementary material, but it would still be useful for the reader to see more of these in the main paper. "*
> >
> > **Response:** We appreciate the reviewer’s valuable feedback on the presentation of key design choices. The losses $\mathcal{L}\_{p}$ and $\mathcal{L}\_{oc}$ are standard in point cloud registration, while $\mathcal{L}\_{f}$ specifically supervises the RIConv++ module in learning the target features. The filtering mechanism is designed to remove edge responses and suppress false positives. To maintain clarity and conciseness within the paper's space constraints, we focus on the core contributions in the main text. We will expand on these design choices in the main text and direct readers to the supplementary materials for a more detailed analysis and experimental validation.
>
>
> > **Question-3:** *" In the ablation study, Table 3: How come the Coord works fairly well? It does not seem to contain that much information intuitively. "*
> >
> > **Response:** While point coordinates may seem limited in expressiveness, they are a fundamental geometric property of point clouds, capturing 3D positions and encoding global shape information. In memory bank-based anomaly detection, coordinates are commonly used for patch matching and similarity comparison. The good performance of "Coord." in Table 3 results from our method’s precise registration, ensuring consistent coordinate correspondence across point clouds, which makes coordinate-based comparisons effective even without additional features.
>
> > **Question-4:** *" What is the intuition behind using $\mathbf{F}''$ and the $\mathcal{P}^{align}$ coordinates as the features in the coreset? Why not other extracted features? "*
> >
> > **Response:** In constructing the coreset, we chose to use feature $\mathbf{F}''$ and aligned coordinates $\mathcal{P}^{align}$ for the following reasons: $\mathbf{F}''$ offers strong rotational invariance and high local resolution, effectively capturing local geometric patterns, while $\mathcal{P}^{align}$ provides 3D position and global shape structure, offering both spatial priors and a structural reference. This combination balances local detail and global consistency, enabling the model to detect subtle local anomalies while maintaining robust geometric coherence.
> >
> > We did not use $\mathbf{F}'$ or $\mathbf{F}$, as they are designed for key point matching to compute the rotation matrix for registration. Their limited receptive fields and weaker semantic representation make them less suitable for capturing the broader geometric context required for effective anomaly detection.

---

> ### Author Response · Authors · 2025-08-06
> **Follow-Upon Rebuttal Response**
>
> Dear Reviewer BTG1,
>
> Thank you for your thoughtful and constructive feedback, as well as your positive assessment of our approach. We have provided detailed responses to your concerns above. We notice that you have not yet acknowledged the rebuttal, submitted your final recommendation, and engaged in further discussion with us. If there are any remaining or unclear issues, we would greatly appreciate the opportunity to clarify or further improve our work. Your feedback is highly valuable, and we are eager to address any remaining points that could help strengthen the paper.
>
> Sincerely,
>
> The Authors

---

### Official Review · Reviewer_PScz · 2025-07-03

**Clarity:** 3
**Significance:** 3
**Originality:** 3
**Rating:** 4
**Confidence:** 4

**Summary:**

This paper tackles the task of 3D point cloud anomaly detection by introducing a registration learner built upon a memory bank. Specifically, the authors train a rotation-invariant feature extractor using point cloud registration as the learning objective, enabling accurate alignment between test samples and templates during inference, while also extracting effective local features. Quantitative results demonstrate the effectiveness of the proposed method.

**Questions:**

1. How long does it take to train the proposed method?

2. What is the inference speed in frames per second (FPS)?

3. Are there any hyperparameters in the proposed method? If so, the authors are expected to conduct a sensitivity analysis to evaluate the robustness of their choices.

4. How does the proposed method perform on noisy point clouds? A discussion or experiment on robustness to noise would strengthen the paper.

5. Do “Coord.” and “Feat.” correspond to L_p, L_oc, and L_f, respectively? If not, an ablation study explicitly analyzing the contributions of these three losses is recommended.

**Ethical Concerns:**

["NO or VERY MINOR ethics concerns only"]

**Final Justification:**

Thanks for the authors' response. After reading the rebuttal as well as the comments from the other fellow reviewers, I believe the paper is a good contribution to the community. As such, I would like to keep my previous positive rating.

**Limitations:**

YES

**Quality:**

3

**Strengths And Weaknesses:**

Strengths:

* The idea of integrating a point cloud registration module into memory-bank-based method is reasonable, which is empirically demonstrated by authors.
* The writing of this paper is clear and easy to read.
* The authors conduct extensive experiments to demonstrate the effectiveness of their method.

Weaknesses:

* I appreciate that authors provide a detailed overview of the proposed method. However, the Figure 2 is somewhat complicated, the authors are expected to simplify it to help readers better understand.
* The authors are expected to provide the equation of L_oc in the main text.

---

> ### Author Rebuttal · Authors · 2025-07-31
>
> #### **Point-by-Point Response to Reviewer PScz**
> We sincerely appreciate your thoughtful and constructive comments, which have greatly enhanced the quality of our manuscript. Below, we provide detailed responses to each of your questions.
>
> > **Question-1:** *" Figure 2 is somewhat complex and should be simplified to improve readability. "*
> >
> > **Response:** Our intention was to align the architectural diagram with the methodology by illustrating the training/inference procedures, as well as the Rotation-Invariant Feature Extractor. We acknowledge the resulting complexity and will simplify Figure 2 by removing unnecessary lines, standardizing dashed styles, and eliminating redundant labels.
>
> > **Question-2:** *" Provide the equation for $\mathcal{L}\_{oc}$ in the main text. "*
> >
> > **Response:** We have provided the equation for $\mathcal{L}\_{oc}$ in the Supplementary Material and will move it to the main text in the revision.
>
> > **Question-3:** *" How long does it take to train the proposed method? "*
> >
> > **Response:**  Our method learns features via a registration task that enforces geometric alignment and multi-scale consistency between source and target point clouds. As it does not rely on category-specific supervision, it is inherently category-agnostic and can be trained jointly across the entire dataset. Under our experimental setup (RTX 3090 GPU, batch size 1), full training takes approximately 27 hours on Anomaly-ShapeNet (40 categories) and about 34 hours on Real3D-AD (12 categories). In contrast, methods like R3D-AD and PO3D-AD require ~1 hour and ~7 hours per category, respectively. While our single training run is longer, it eliminates the need for category-wise training, resulting in significantly greater overall efficiency.
>
> > **Question-4:** *" What is the inference speed in frames per second (FPS)? "*
> >
> > **Response:** Thank you for the suggestion. While FPS is less commonly used for point cloud anomaly detection due to variable point counts and the absence of a clear “frame” definition, we report **average inference time per sample** as a practical alternative. Under our setup (RTX 3090 GPU, batch size 1), our method takes an average of 2.53 seconds to process one sample on the Real3D-AD dataset, outperforming other memory bank-based methods, as shown in **Table 1** below.
> >
> > **Table 1:** Comparison of Inference Speed.
> > | Method | average inference time per sample (seconds) |
> > | :---: | :---: |
> > | RegAD | 7.71 |
> > | M3DM | 6.43 |
> > | ISMP | 4.37 |
> > | **Ours** | **2.53** |
>
> > **Question-5:** *" Does the proposed method involve hyperparameters, and has a sensitivity analysis been conducted to assess their robustness? "*
> >
> > **Response:** Yes, our method involves a key hyperparameter: the neighborhood size in RIConv++. To assess its impact, we conducted a sensitivity analysis on the Real3D-AD dataset, as shown in Figure 1 of the supplementary material. The results indicate that smaller neighborhood sizes limit contextual information, while larger sizes may weaken local feature discrimination by aggregating dissimilar points. Despite these effects, our method performs consistently well across a range of values, demonstrating robustness to this hyperparameter.
>
> > **Question-6:** *" How does the proposed method perform on noisy point clouds? "*
> >
> > **Response:** We appreciate the reviewer’s insightful comment. Like most models, ours is sensitive to unexpected noise during testing. To assess robustness, we conducted experiments on the Real3D-AD dataset under two settings. In Setting 1, Gaussian noise (SD = 0.005) was added during training, with various noise levels applied during testing. In Setting 2, noise was added only during testing. As shown in **Tables 2 and 3** below, the model without noise augmentation (Setting 2) shows a noticeable performance drop at higher noise levels (SD ≥ 0.003), indicating sensitivity to input perturbations. However, with noise-based augmentation during training (Setting 1), the model maintains stable performance across different noise levels. These results suggest that, with minor fine-tuning, our method can effectively handle noisy point clouds.
>
> > **Table 2:** Results of Setting-1
> > | noise | O-AUROC | P-AUROC |
> > | :---: | :---: | :---: |
> > | clean | 78.0 | 87.8 |
> > | SD=0.001 | 76.7 | 87.5 |
> > | SD=0.003 | 77.7 | 86.4 |
> > | SD=0.005 | 76.8 | 83.1 |
> >
> > **Table 3:** Results of Setting-2
> > | noise | O-AUROC | P-AUROC |
> > | :---: | :---: | :---: |
> > | clean | 78.0 | 87.8 |
> > | SD=0.001 | 76.9 | 86.5 |
> > | SD=0.003 | 63.0 | 69.1 |
> > | SD=0.005 | 56.6 | 56.5 |
>
> > **Question-7:** *" Do "Coord." and "Feat." correspond to $\mathcal{L}\_{p}$, $\mathcal{L}\_{oc}$, and $\mathcal{L}\_{f}$, respectively? If not, an ablation study explicitly analyzing the contributions of these three losses is recommended. "*
> >
> > **Response:** Thank you for the question. "Coord." and "Feat." refer to the registered point coordinates and the learned local features, respectively, but they do not directly correspond to $\mathcal{L}\_{p}$, $\mathcal{L}\_{oc}$, and $\mathcal{L}\_{f}$. To clarify their contributions, we conducted an ablation study on these three loss terms using the Real3D-AD dataset, with results shown in **Table 4** below.
>
> > The analysis shows that each loss serves a distinct role:
>
> > 1. $\mathcal{L}\_{oc}$ is the most critical for accurate registration, enabling the model to align structurally consistent local patches even under partial overlap or sparsity. Removing it severely degrades performance.
>
> > 2. $\mathcal{L}\_{f}$ enhances the rotation invariance and local discriminability of features, directly improving anomaly localization accuracy.
>
> > 3. $\mathcal{L}\_{p}$ has a smaller impact on detection accuracy but is important for efficient point matching. Without it, inference time increases significantly due to many spurious matches.
>
> > In summary, $\mathcal{L}\_{oc}$ and $\mathcal{L}\_{f}$ primarily support accurate feature learning and detection, while $\mathcal{L}\_{p}$ improves inference efficiency. All three contribute to balancing performance and computational cost. We will include these discussions in the revision.
>
> > **Table 4:** Ablation study on loss components.
> > | loss components | O-AUROC | P-AUROC |
> > | :---: | :---: | :---: |
> > | w/o $\mathcal{L}\_{oc}$ | 58.1 | 68.1 |
> > | w/o $\mathcal{L}\_{p}$ | 76.9 | 87.0 |
> > | w/o $\mathcal{L}\_{f}$ | 73.4 | 86.0 |
> > | $\mathcal{L}\_{oc}$ + $\mathcal{L}\_{p}$ + $\mathcal{L}\_{f}$ | 78.0 | 87.8 |

---

> ### Author Response · Authors · 2025-08-07
> **Final Rating Reminder**
>
> Dear Reviewer PScz,
>
> We sincerely thank you once again for your constructive feedback and positive evaluation of our work. As the review process nears completion, we would greatly appreciate it if you could kindly submit your final rating at your earliest convenience.
>
> Sincerely,
>
> The Authors

---

### Author Response · Authors · 2025-08-05

Dear Reviewers,

We sincerely appreciate the thoughtful and constructive feedback you have provided during the review process. We have carefully addressed all raised concerns and remain happy to clarify any further questions if needed. As the review process approaches its deadline, we are grateful for your continued engagement and contribution.

Sincerely,

The Authors

---

### Author Response · Authors · 2025-08-06

Dear Reviewers,

With the deadline approaching, we hope our previous responses have sufficiently addressed your concerns. If there are any remaining questions, we would be glad to clarify them further. We believe that an open and transparent discussion will not only help elevate our method but also benefit the broader research community.

Sincerely,

The Authors

---

### Note · Authors · 2025-08-12

We sincerely thank the AC and reviewers for their thoughtful and constructive feedback.

In our original submission, the reviewers recognized several key strengths of our work:

1. **Novelty.** This is the first to integrate point cloud registration into a memory-based 3D anomaly detection framework, leveraging registration learning to obtain rotation-invariant and highly discriminative features, which is both intuitive and innovative.

2. **Importance.** Our work addresses the critical challenge of effective feature learning in 3D point cloud anomaly detection, offering both strong practical and academic value, and making a meaningful contribution to the field.

3. **Effectiveness.** Extensive experiments robustly validate the effectiveness and advantages of our proposed method.

4. **Clarity.** The paper is clearly written, well structured, and easy to follow.

The types of comments raised by the reviewers can be grouped into 5 categories:

1. Clarity issues in certain figures and tables.

2. Requests for additional experimental details, including training time, inference speed, robustness to noise, and other relevant aspects.

3. Calls for more comprehensive ablation studies.

4. Requests for discussion on the method’s applicability to other scenarios.

5. Suggestions for improving the writing.

These comments focused on improving the clarity of the paper’s figures and tables, providing more comprehensive experimental details, validation, and extensions of our Reg2Inv, as well as enhancing the writing quality. **Importantly, they did not challenge the core innovation, contribution, or validity of our approach.** All have been thoroughly addressed during the rebuttal period, with additional experiments and clarifications provided. **Reviewers have acknowledged these improvements.**

We believe the revised work presents a **novel, impactful, and rigorously validated** solution for robust 3D point cloud anomaly detection.

---

### Decision · Program_Chairs · 2025-09-17

**Decision:**

Accept (poster)

**Comment:**

The main idea of the paper is to integrate the objectives of point-cloud registration and memory-bank-based 3D anomaly detection. The proposed framework utilizes registration learning to derive rotation-invariant and locally discriminative features.

The paper initially received mixed reviews. It had one rating of "accept," one rating of "borderline accept," and two ratings of "borderline reject." The main issues noted in the reviews include the clarity of figures and tables, the length of training time and inference speed, and the need for analysis of sensitivity and robustness. Additionally, concerns were raised about the weak ablation study, a lack of intuition behind some design choices, and insufficient detailed analysis of the experiments.

In response, the authors provided explanations and additional experiments to address these issues. After the rebuttal, one reviewer who had initially rated the paper as "borderline reject" upgraded the score to "borderline accept." Another reviewer commented, "I believe the paper is a good contribution to the community." None of the reviewers expressed any further concerns. As a result, the overall ratings shifted toward a more positive outlook.

The area chair does not see any significant flaws and considers the remaining issues, such as visualization and analysis, to be minor and easily addressable. The area chair concurs with the reviewers' assessments and recommends accepting the paper. Additionally, the area chair has discussed the evaluations and recommendations regarding this paper with the senior area chair to ensure that all perspectives have been fairly considered in the decision-making process.